# Ecosystem impacts of marine heat waves in the Northeast Pacific

Abigale M. Wyatt[1], Laure Resplandy[1,2], Adrian Marchetti[3]

[1]Department of Geosciences, Princeton University, Princeton, NJ, USA

[2]High Meadows Environmental Institute, Princeton University, Princeton, NJ, USA

5    [3]Earth, Marine and Environmental Sciences, University of North Carolina, Chapel Hill, NC, USA

*Correspondence to*: Abigale Wyatt, awyatt@princeton.edu

**Abstract.** Marine heatwaves (MHWs) are a recurrent phenomenon in the Northeast Pacific that impact regional ecosystems and are expected to intensify in the future. Prior work showed that these events, including the 2014–2015 "warm blob," are associated with widespread surface nutrient declines in the subpolar Alaska Gyre (AG) and the North Pacific Transition Zone (NPTZ), but reduced chlorophyll concentrations in the NPTZ only. Here we explain the contrast between these two regions using a global ocean-biogeochemical model (MOM6-COBALT) with Argo float and ship-based observations to investigate how MHWs influence marine productivity. We find that phytoplankton and zooplankton production respond relatively modestly to MHWs in both regions. However, differences in the response to seasonal iron and nitrogen limitation between large (>10 μm) and small (<10 μm) phytoplankton size classes explain the differences in ecosystem response to MHWs across the two biomes. During MHWs, reduced nutrient supply limits large phytoplankton production in the NPTZ (-13 % annually), but has a limited impact on the already iron-limited large phytoplankton population in the AG (-2 %). In contrast, MHWs yield a springtime increase in small phytoplankton in both regions due to shallower mixed layers and weaker light limitation. These modest changes are in apparent contradiction with prior estimates suggesting a collapse in net community production during the warm blob. We show, however, that 70% of the decline in net community production previously calculated from nitrate Argo data can be attributed to artifacts in the method and that only 30% can be attributed to interannual variability, in line with our model-based results. Although modest, the primary production anomalies associated with MHWs modify the phytoplankton size distribution, resulting in a significant shift towards small phytoplankton production (i.e. lower large to small phytoplankton ratio) and reduced secondary and export production, especially in the NPTZ.

## 1 Introduction

30    Marine heat waves (MHW) are a recurring phenomenon in the Northeast Pacific, with nine events on record since 1958 (Xu et al. 2021). The largest such event, which occurred during the satellite chlorophyll era, was a persistent marine heat wave known as the "warm blob" that appeared in 2014 and 2015 and was characterized by a greater than 2° C surface temperature anomaly in the Northeast Pacific (Freeland and Whitney 2014; Bond et al. 2015; Di Lorenzo and Mantua 2016). The 2014–2015 marine heat wave broadly influenced ecosystems in the northeast Pacific Ocean with a shift in marine species' geographical distribution and anomalous appearances of fish species outside of their known range across the northeast Pacific (See Bond et al. 2015) with some effects persistent or permanent (Suryan et

al. 2021). In situ observations indicate that the warm blob particularly affected ecosystems in two regions: the subpolar Alaska Gyre (AG) and the Northeast Pacific Transition Zone (NPTZ roughly between 30º N to 45º N), i.e. the region of strong chlorophyll and nitrate gradient that demarcates the boundary between the AG and the eastern subtropical gyre. Major impacts of this warm blob included a ~35 % decrease in satellite surface chlorophyll in the NPTZ (Whitney et al. 2015), a reduction in surface nitrate concentrations and phytoplankton biomass, and an increase in cyanobacteria dominance along the subarctic transect Line P which samples both the NPTZ and AG regions (near 50º N, Peña et al. 2019). Further, estimates of net primary productivity suggested there was an ecosystem collapse in the second year of the warm event near Ocean Station Papa in the AG (OSP, 50.1° N, 144.9° W, Bif et al. 2019b), despite a lack of satellite surface chlorophyll anomaly in this region.

Prior work offered a bottom-up explanation for the chlorophyll anomalies observed in the NPTZ during MHW, noting that the 2014–2015 heat wave was associated with decreased winds that reduced nitrate concentrations and inhibited primary production (Whitney 2015). This bottom-up explanation does not explain why the decrease in chlorophyll was highly localized (confined to the NPTZ) while anomalously low nitrate concentrations extended 600 km north (into the AG) of any significant chlorophyll anomalies (Peña et al. 2019). In addition, surface chlorophyll alone provides little information on food web changes or how marine heat waves influence secondary production and marine biogeography. Finally, it is unclear to what extent the observed anomalies in nitrate and chlorophyll are unique to the "warm blob" or typical of the MHWs in this area.

The AG and NPTZ are distinct ecological biomes. The AG is a high nutrient, low chlorophyll (HNLC) region, characterized by high nitrate concentrations, but moderate primary production throughout the year due to iron limitation that prevents the development of a strong spring bloom (Martin and Fitzwater 1988; Harrison 2002; Boyd et al. 2004; Peña and Varela 2007). In contrast, the NPTZ is a region characterized by strong seasonality in nitrate and chlorophyll due to the seasonal biological consumption and the Ekman transport of nutrients ( Chai et al. 2003; Polovina et al. 2008; Ayers and Lozier 2010). As a result, the NPTZ evolves from a subpolar-like, iron-limited biome when nitrate is abundant in spring to a nitrate-depleted, subtropical-like biome in summer, with the position of the chlorophyll front associated with the bloom (2 mg m$^{-3}$ chlorophyll contour) shifting ~10º northward in summer from its southernmost position in winter (30º to 40º N, Bograd, et al. 2004, Glover et al., 1994).

In this study, we examine the ecosystem response to the nine MHWs that were recorded since 1958 in the AG and NPTZ biomes. Using a combination of observations and ocean biophysical model results, we first characterize MHWs in section 3.1. Then in section 3.2, we examine the extent of nitrate depletion during MHWs and show that the boundary between the subpolar HNLC region and the NPTZ shifts during these events, expanding the region of nitrate depletion. We then analyze the biological response to MHWs in the NPTZ and the contrasting response in the AG (sections 3.3 and 3.4), with particular emphasis on the responses of the two phytoplankton size classes. Our results indicate that during MHWs, though the chlorophyll anomaly is confined to the NPTZ, both regions exhibit a shift in

the phytoplankton assemblage toward the smaller size class, resulting in the reduction of secondary and export production.


## 2 Methods and datasets

### 2.1 Definition of northeast Pacific marine heatwaves

Following the method of Xu et al. 2021, we calculate the area mean sea surface temperature anomalies (SSTa)
relative to the climatology of the region $35^\circ$ to $46^\circ$ N, $150^\circ$ to $135^\circ$ W using the monthly data from 1958 – 2020 of the Extended Reconstruction SST dataset (ERSSTv4, Huang et al., 2015). Northeast Pacific marine heatwaves are defined as periods when the monthly deviation relative to the climatology exceeds 1 standard deviation for 5 months or more. The same method is used to detect marine heatwaves in the ocean model (see Sect. 2.3 for model details). We define heatwaves considering their impact on the spring–summer blooming season. For example, what we refer to as the
"year 1990 heatwave" started in November 1989 and ended in March 1990, thus impacting the ecosystem in 1990. In both ERSST and the model, the "marine heatwaves" or "warm years" selected using these criteria are 1962, 1963, 1965, 1990, 1991, 2005, 2014, 2015 and 2019 similar to what was found by Xu and coauthors (Fig. 1).

### 2.2 Composite anomalies and statistical analysis


We compute the composite of the nine MHW events to evaluate the impact of heatwaves on marine ecosystems. examining SST, mixed layer depth (MLD), surface nutrients and 6 ecosystem variables (chlorophyll, large phytoplankton production, small phytoplankton production, ratio of large to small phytoplankton production, zooplankton production, and export production). To calculate the MHW composite, we remove the 1958 – 2020 linear
warming trend at each model grid point. The climatology of all years was calculated at each spatial point and removed from the 9 selected MHW years to get the annual anomalies of each MHW year. The 9 MHW years were then averaged together to get a single, composite MHW year. To quantify the size of the perturbations caused by MHWs, we compared the magnitude of the MHW anomalies to the variability during non-MHW years calculated as the average monthly standard deviation. We focus on two subregions representative of the NPTZ ($39^\circ$ to $45^\circ$ N and $160^\circ$ to $135^\circ$
W) and the HNLC Alaska Gyre ($48^\circ$ to $54^\circ$ N and $160^\circ$ to $145^\circ$ W)

To test whether the spatially averaged MHW anomalies (N=9) differed significantly from the non-MHW years (N=53), the 6 ecosystem variables were compared across the two datasets using the two-variable Kolmogorov-Smirnov (K-S) test. This test is a suitable choice as it does not require a normal probability distribution or equal
variance of the two datasets. A threshold p-value < 0.05, is used throughout this manuscript to indicate "significant" differences, indicating that changes in that variable are attributable to MHWs. For annual values, production variables were annually integrated, and chlorophyll was annually averaged. For seasonal values, the appropriate month(s) were selected and averaged. As we used annual or seasonal mean data, we assume that autocorrelation is negligible and calculate significance based on the total number of years in the time series.


### 2.3 Line P data processing

We use Line P observations of temperature, salinity, nitrate and Chl-a available online (downloaded from www.waterproperties.ca/linep on Mar 19, 2021). Data from the two summertime cruises, May & June and Aug/Sep,
were averaged at each of the 26 stations from 2007 to 2020. The Jan/Feb cruise data were not used as we focused on the period of seasonal nitrate depletion. For comparison, the model results were sampled at the same station locations, averaged across June, July and August each year to obtain a summer mean.

### 2.4 Ocean biogeochemical model (MOM6-COBALT)


This study uses the biophysical ocean model described in Liao et al. 2020. This model configuration uses the fourth generation global ocean/sea ice model OM4p5 developed at the Geophysical Fluid Dynamics Laboratory, consisting of the Modular Ocean Model version 6 (MOM6) and the Sea Ice Simulator version 2 (SIS2, Adcroft et al. 2019). The physical ocean circulation model has a nominal 0.5° x 0.5° resolution in the horizontal and 75 hybrid depth-isopycnal
z* layers in the vertical. The physical model is coupled with the biogeochemical model Carbon, Ocean Biogeochemistry and Lower Trophics v.2 (COBALTv.2) that simulates a nitrogen-based ecosystem with 33 biochemical tracers and 13 food web components (Stock et al 2014, 2020). These components include three phytoplankton size classes: large (>10 $\mu$m), small (<10 $\mu$m), and nitrogen-fixing diazotrophs. Phytoplankton growth is explicitly modeled as size-dependent functions of light, temperature, and nutrient limitations (nitrate, ammonia,
phosphate, etc.). Small phytoplankton are simulated to be efficient nutrient and light harvesters (Munk and Riley 1952; Geider et al. 1997) in contrast to large phytoplankton which are parameterized to grow quickly in response to abundant nutrients. For macronutrients (e.g. nitrogen, phosphate), limitation factors are calculated using saturating kinetics while for iron, an internal iron deficiency term is calculated based on an internal cell quota (see supplementary materials equations 1–5 in Stock 2014 for details). These limitation factors are output from the model as a number
between zero and one, with zero indicating complete limitation, i.e. no phytoplankton growth. The nutrient with the lowest value is considered the limiting nutrient (Droop 1983). Notably, in the study regions iron is only a limiting nutrient for large phytoplankton. The model also includes three zooplankton size classes of which large (>2000 $\mu$m) and medium (200 to 2000 $\mu$m) make up the mesozooplankton pool with a third, separate small zooplankton class (<200 $\mu$m) all of which consume phytoplankton using size-related predator-prey relationships. These nitrogen-based
biological tracers are assumed to maintain a stoichiometric relationship with carbon in accordance with the Redfield ratio, 106C:16N. Chlorophyll is calculated from phytoplankton biomass using a Chl:C ratio that depends on ambient light, temperature, iron availability and size-class specific nutrient limitation and maximum photosynthetic rates (Geider et al. 1997; Stock et al. 2014, 2020).

The model was spun-up from rest using three repetitions of the 1958 to 1985 Japanese atmospheric reanalysis v1.4 (JRA55do v1.4, Tsujino et al. 2018) for a total of 81 years. Initial nutrient, temperature and salinity fields are from the

2013 World Ocean Atlas (WOA, Boyer et al. 2013). Dissolved inorganic carbon (DIC) and alkalinity are from the global ocean data analysis project v2 climatologies (GLODAPv2, Olsen et al. 2016) with DIC corrected to 1958 using anthropogenic carbon concentrations from Khatiwala et al. (2013). Initial states of the remaining tracers (e.g. Chl, biomass, etc) were taken from a long, preindustrial control run from the GFDL Earth system model ESM2M-COBALT (Dunne et al. 2012). The model was then run from 1958 to 2019 using the JRA v1.4 forcing and river nutrient fluxes taken from the Global NEWS climatology (Seitzinger et al. 2010).

**2.5 Size-fractionated Chlorophyll *a* concentration at OSP**

Discrete summertime measurements of mixed layer, size-fractionated Chlorophyll *a* (Chl *a*) concentrations at OSP were obtained through collection of 300 mL of seawater from a Rosette system during Line P cruises in June of 2000, 2001, 2008, 2013, 2015 and 2018. The 2015 sample was taken during the "warm blob," while the 2013 sample was collected following the Mt. Pavlof eruption (Waythomas et al. 2014). Seawater was vacuum filtered through a 5 µm pore-sized polycarbonate filter, and the filtrate was passed through a GF/F filter (0.7 µm nominal porosity) set up in series. Filters were frozen at -80º C until analysis. Chl *a* extraction was performed using 90 % acetone or ethanol at -20º C overnight and concentrations were determined fluorometrically using a Turner Designs 10-AU fluorometer (Brand et al. 1981).

Phytoplankton in the study regions are classified into two allometric classes. Small phytoplankton (< 5 $\mu$m vs. < 10 $\mu$m in model), primarily made up of cyanobacteria (e.g. *Synechococcus*) and nanoflagellates such as chlorophytes and haptophytes, comprise the majority of the biomass in both regions (Boyd and Harrison 1999). Large phytoplankton (> 5 $\mu$m vs. > 10 $\mu$m in model), primarily made up of dinoflagellates and diatoms, have a stronger correlation to particulate export production (Buesseler 1998), are subject to iron limitation inside the AG.

**2.6 Argo floats & other datasets**

This study makes use of the 2008–2018 series of BGC-Argo floats with nitrate sensors deployed near OSP (Fig. 2). Specifically, we replicated the analyses of Plant et al. 2016 as updated in Bif et al. 2019a to evaluate net community production (NCP) from BGC-Argo float nitrate data then compare NCP estimates derived from nitrate concentrations in the WOA climatology and the MOM6-COBALT model (See appendix A). Following the quality control analyses of those studies, selected profiles from floats 5903405, 5903891 and 5903714 were dropped due to inconsistencies in the nitrate data (See Bif and Hansell 2019a). Satellite chlorophyll observations (1997 to 2020) are from the GlobColour dataset (http://globcolour.info) which has been developed, validated, and distributed by ACRI-ST, France (Maritorena et al. 2010).

**3 Results**

**3.1 Characterizing marine heat waves in observations and MOM6-COBALT ocean model**

Marine heat waves show systematically high SST over a relatively broad area of the northeast Pacific that extends from 35º N to 55º N and from 170º E to the North American coast in both observations and the MOM6-COBALT model (up to +1ºC in average across the nine events, Fig. 1c,d). Surface Chl, in contrast, exhibits more spatial heterogeneity, with a strong decline in the NPTZ (-0.05 mg m$^{-3}$) and a mild increase further north in the AG (+0.02 mg m$^{-3}$ around station OSP, Fig. 1d & e).


We use the observations from 6 Argo floats that sampled the AG region around OSP between the years 2008 and 2019 to characterize interannual variability in the region (Fig. 2). These data show the strong signal associated with the 2014–2015 warm event, colloquially termed the "warm blob", including summer surface temperatures above 15º C and surface nitrate concentrations below 6 $\mu$mol kg$^{-1}$ (Fig. 3 a, c, e). Using the ocean model sampled along the floats'

trajectories yields similar features, with modeled temperatures exceeding 14º C and nitrate concentrations dropping to <3 $\mu$mol kg$^{-1}$ during the warm blob period (Fig. 3 b, d, f). It is worth noting that in both the observed and modeled profiles, large changes in temperature, nitrate and to a lesser extent salinity are apparent at depth (>100 m) in early 2015 (see also sampled WOA profiles in Fig S2). These subsurface changes were sampled by a single float (#5904125, brown, Fig. 2) and likely indicate sampling of a different water mass with a shallower thermocline and nitracline, in

this case the inner AG (See Sect. 4.2). Regardless, these data support the bottom-up explanation of Whitney (2015) that posited reduced surface nutrient concentrations as a driver of reduced primary production and chlorophyll concentrations during the "warm blob".

We can further observe this impact of the 2014–2015 marine heat wave on nitrate and chlorophyll concentrations

using 2007–2020 summer cruise data (June – September) from the Canadian Line P program, which sampled from the coast of British Columbia to OSP (yellow dots, Fig. 1). Fig. 4 shows a strong signal during the 2014–2015 "warm blob" along Line P, characterized by higher SSTs (+2.5º C) and lower SSS (~0.2 PSU) between 130º W and 140º W (Fig. 4). During this period, observed chlorophyll data reached concentrations below 0.3 mg m$^{-3}$ (Fig. 4g) while nitrate concentrations are near-zero west of P4 (P4–P20, Fig. 4a). We sampled the model results at Line P stations and found

similar results, including SST (+2º C) and salinity (-0.1 PSU) anomalies during the "warm blob", and despite a model bias toward lower climatological surface nitrate in this region (Fig. 2), the nitrate anomaly associated with the marine heat wave is still well simulated (-2 $\mu$M, Fig. 4b). The observed chlorophyll anomaly is difficult to characterize due to the patchiness of the chlorophyll field, however, the simulated chlorophyll in the model strongly suggests a decline (<0.3 mg m$^{-3}$) during the 2014–2015 period.


**3.2 Northward expansion of nitrate-depleted region in response to marine heatwaves**

The northeast Pacific is characterized by three regions: the nitrate-rich HNLC AG, the nitrate-depleted subtropical gyre, and the NPTZ region in between. Climatologically, WOA observations show that in winter, the nitrate-depleted region (identified here with surface $NO_3$ <2 $\mu$M) extends from ~35º N on the western side of the region to ~45º N in the east near the North American coast (Fig. 5a, blue line). By the end of summer (September, green line), biological consumption has expanded the nitrate-depleted region, shifting the 2 $\mu$M contour by about 2 to 5º northward between 180º and 140º W and by about 10º east of 135º W and along the North American coast. This seasonal displacement of the nitrate front is also captured in the MOM6-COBALT climatology, with an ~8º northward shift in the western region and a similar 10º northward shift along the North American coast (Fig. 5b). The large-scale north-south nitrate gradient is, however, more intense in the model, with an approximately -2 $\mu$M annual mean nitrate bias in the transition zone and a +2 $\mu$M bias in the northwest AG (Fig. S1). Here we combine in situ observations with the results of the MOM6-COBALT ocean biogeochemical model and show that these warm events also systematically expand the spatial extent of nitrate depletion northward.

Examining the nine warm events (1962, 1963, 1965, 1990, 1991, 2005, 2014, 2015, 2019), we find that there is generally an expansion of the nitrate-depleted region northward into the NPTZ during warm events (Fig. 5b). Compared to the climatological 2 $\mu$M nitrate contour (solid lines), the 2 $\mu$M nitrate contour during marine heatwaves (thin dashed lines) is located ~2º further north on average in February and ~1º north on average in September (Fig. 5b), with the model suggesting that the nitrate contour shift is most consistent in the NPTZ. The WOA does not provide interannual information that we can use to evaluate the response to marine heat waves, but we can use observations from the sampling program at Line P (yellow dots), which intersects the 2 $\mu$M nitrate contour (i.e. transition between the nitrate-depleted and the nitrate-replete regions) in summer to examine its response to the 2014–2015 event.

The Line P program's June and August cruises sample three regimes (Fig. 4): the highly variable but generally nutrient-rich near-shore region (>10 $\mu$M at ~125º W), followed by the seasonally nitrate-depleted region that extends to roughly 130º W, before reaching the third region characterized by moderate to high nitrate concentrations (>5 $\mu$M) in the iron-limited AG. Ship-based Line P observations show that the high nitrate concentrations along the coast and in the AG are co-located with colder sea surface temperatures (SST < 12º C), and higher chlorophyll concentrations (> 0.5 mg m$^{-3}$) in comparison to those observed in the nutrient-depleted region (Fig. 4a). There is a gradient in salinity across the region, with the highest salinity in the west near OSP (~32.4 PSU) and fresher water near shore in the east (<32 PSU, Fig. 4c). These observed patterns are replicated in the MOM6-COBALT model, including the east-west contrasts in surface nitrate, SST, SSS and chlorophyll between the coastal region, the nitrate-depleted region and the subpolar gyre (Fig. 4). However, we note that the modeled surface nitrate concentration is generally lower in comparison to the Line P data, with maximum values rarely exceeding 8 $\mu$M versus 15 $\mu$M in the observations (Fig. 3a–b) consistent with the annual mean nitrate bias mentioned above.

The Line P data support the model result and show an expansion of the nitrate-depleted region during the 2014–2015 "warm blob" (Fig. 4), leading to a westward shift of the 2 $\mu$M contour to 140º W in 2014 (vs a location of ~130º W in

the other years). In the model, this westward shift of the nitrate contour is overestimated, extending past 140° W. However, in both the observations and model this implies that nitrate becomes depleted inside the climatological boundary of the HNLC AG. The HNLC region can therefore be considered to contract while the nitrate-depleted region expands.

**3.3 Reduced ecosystem production and export in NPTZ**

To understand the biological impacts of marine heat waves, we examine the composite of the nine simulated warm events. As expected from observations (Whitney 2015; Le et al. 2019), the model simulates the greatest biological anomalies in the NPTZ, including a negative chlorophyll anomaly (-0.03 mg m$^{-3}$, $p < 0.05$, Fig. 6a) comparable with

satellite observations (Fig. 1e). This chlorophyll anomaly is spatially co-located with a shallow winter mixed layer anomaly (-10 m, $p < 0.05$, Fig. 6b) that reduced winter surface nitrate (-2 $\mu$M, $p < 0.05$, Fig. 6c) but has little effect on winter iron concentrations (-0.1 nM, $p > 0.05$, Fig. 6d). The low winter supply of nutrients during these events inhibits the annual production of both large (-8 mmol C m$^{-2}$ Fig. 6e) and small (-6 mmol C m$^{-2}$ Fig. 6f) phytoplankton inside the NPTZ ($p < 0.05$ for both). These negative anomalies in phytoplankton production propagate through the

food web, leading to a drop in simulated zooplankton production of all three size classes (small, medium and large) and thus significantly low total annual secondary production (-1 mmol m$^{-2}$ d$^{-1}$, $p < 0.05$, Fig. 6g). Similarly, particle export production, which includes zooplankton egestion and phytoplankton aggregation, also exhibits a negative production anomaly concentrated in the NPTZ (-0.5 mmol m$^{-2}$ d$^{-1}$, $p < 0.05$, Fig. 6h). Although the small phytoplankton and zooplankton MHW production anomalies are relatively small in magnitude and within one standard deviation of

the model interannual variability ($<1\ \sigma$, Fig. 7b), they are statistically different from the mean state (K-S test, all p-values $< 0.05$). We note, the most substantial response in the composite was the reduction in the ratio of large to small phytoplankton production ($p \ll 0.01$), the magnitude of of which exceeds the interannual standard deviation ($1\ \sigma$, Fig. 7b). Individually, the annual response MHW events varies from an intense signal in 1965 (e.g. phytoplankton, zooplankton and export production anomalies $>2\ \sigma$) to a near-zero perturbation in 1990.


While MHWs yield negative anomalies in annual primary and secondary production in the NPTZ (Fig. 6e–h), the effect varies seasonally, following the ecosystem size-class succession. Climatologically, the NPTZ in the model is characterized by a winter supply of nutrients supporting a modest spring bloom of large phytoplankton that peaks in April (13.5 mmol C m$^{-2}$ d$^{-1}$, Fig. S4e) followed by a much larger peak in small phytoplankton production in June (47

mmol C m$^{-2}$ d$^{-1}$, Fig. S4f) that dominates total primary production. Seasonal chlorophyll largely follows the large phytoplankton production due to a higher simulated Chl:C ratio for large phytoplankton (0.022 vs. 0.014) in this region (Geider et al. 1997; Stock et al. 2020). Thus, chlorophyll peaks in April ($>0.6$ mg chlorophyll m$^{-3}$) with more modest values during the small phytoplankton peak in June (0.2 mg chlorophyll m$^{-3}$). Zooplankton production also follows a size-based progression, with medium-size zooplankton, the primary consumer of large phytoplankton, peaking first in

May (0.8 mmol m$^{-2}$ d$^{-1}$, Fig. S4g), followed by small zooplankton peaking in May/June (1.4 mmol m$^{-2}$ d$^{-1}$) then large

zooplankton, which consume both large phytoplankton and medium zooplankton, peaking last in June (0.2 mmol m$^{-2}$ d$^{-1}$, Fig. S4f).

Marine heat waves modulate this climatological progression of the ecosystem in the NPTZ (Fig. 8). The model suggests that marine heat waves promote the growth of small phytoplankton and small to medium-sized zooplankton in early spring before declining in summer–fall (Fig. 8e,f). This enhanced growth in the model is due to the shallower mixed layer in winter and early spring (-10 m, p < 0.05, Fig. 8b) that relieves light limitation and spurs early small phytoplankton and subsequent zooplankton production (Fig. 8e, 8f) but has little impact on the spring large phytoplankton production (Fig, 8e). Iron limitation, which dominates January–April, is not significantly impacted during MHWs (Fig. 8h), however, the onset of nitrogen limitation which occurs when the nitrogen limitation factor (dotted red line) intersects and the iron limitation factor (dotted blue line), happens nearly a month earlier (early April) than the climatology (solid lines, late April). Further, the nitrogen limitation factor during this period is significantly lower (-0.06, p < 0.05, ~25% of the seasonal signal). Both size classes are limited by the reduced pool of nitrate, with negative anomalies in June (-5 mmol m$^{-2}$ d$^{-1}$ for small phytoplankton; -2 mmol m$^{-2}$ d$^{-1}$ for large phytoplankton; -2.5 mmol C m$^{-2}$ d$^{-1}$ for total zooplankton production, Fig. 8e-f; p < 0.05 for all) when nitrate approaches depletion (Fig. S4).

Even with the small increase in early spring small phytoplankton production, the annual mean surface chlorophyll anomaly in the model is significantly negative (-0.03 mg m$^{-3}$, p < 0.05, Fig. 6a) in agreement with satellite observations (Fig. 1e). This slight increase in small phytoplankton production in early spring is only slightly apparent in both modeled and observed chlorophyll (red and green lines, Fig. 8a) as the impact on surface chlorophyll is small. This is again explained by the higher simulated Chl:C ratio of large phytoplankton compared to small phytoplankton which controls the overall response of chlorophyll to marine heat waves in this region. Indeed, a 4 % decrease in total phytoplankton production yields a 11 % decline in Chl, more closely resembling the decrease in large phytoplankton production (-12 %) than the decreased production of the more dominant but less Chl-dense small phytoplankton (-2 %). This model result is consistent with the decrease in chlorophyll captured by satellite observations.

We examine the changes in phytoplankton assemblage across the NPTZ, using the normalized probability density functions of summer chlorophyll concentrations (Fig. 10). In the NPTZ (Fig. 10c, d), the distribution of phytoplankton chlorophyll concentrations for both size classes is bimodal, with one mode consistent with high chlorophyll concentrations typically found in the subpolar AG (large phytoplankton chlorophyll peak centered at 0.28 mg m$^{-3}$; small phytoplankton chlorophyll peak centered at 0.25 mg m$^{-3}$, similar to the AG distribution shown in Fig. 10a–b), and one mode consistent with low chlorophyll concentrations typically found in the subtropical gyre (Large phytoplankton chlorophyll peak centered at 0.02 mg m$^{-3}$; Small phytoplankton chlorophyll peak centered at 0.06 mg m$^{-3}$). During marine heat waves, the chlorophyll distribution in the NPTZ exhibits a significant shift towards lower chlorophyll concentrations, though the shift is greater for large phytoplankton (shift of -0.05 mg m$^{-3}$ in the mean chlorophyll concentration; p << 0.01) than for smaller phytoplankton (-0.02 mg m$^{-3}$ in the mean chlorophyll

concentration; p << 0.01). The model suggests that climatologically, 31 % of the NPTZ area has chlorophyll concentrations < 0.15 mg m$^{-3}$ for the large phytoplankton size-class, but that the proportion of the NPTZ with such low chlorophyll concentrations increases to 41 % during marine heat waves (Fig. 10, Fig. S6). Similarly, the proportion of the NPTZ with low small phytoplankton chlorophyll concentrations (chlorophyll < 0.15 mg m$^{-3}$) increases from 28 % in the climatological state to 38 % during marine heat waves. In both cases, this shift is consistent with a decrease in the high-chlorophyll mode and an increase in the low-chlorophyll mode, and consistent with the decline in satellite chlorophyll observed in this region.

### 3.4 Modulated response in the Alaska Gyre

North of the NPTZ, in the AG, the biological impact of marine heatwaves is less prominent, with the model suggesting that decrease in annual large phytoplankton production is compensated by an increase in small phytoplankton production. Generally, the drivers of the ecosystem response to marine heat waves in the AG resemble the response simulated in the NPTZ (see Sect. 3.3) but the balance between the light-driven increase in small phytoplankton and nutrient-driven reduction in large phytoplankton is different. Specifically, we find that shallow mixed layers reduce light limitation during marine heat waves and trigger an increase in spring small phytoplankton production (+2 mmol C m$^{-3}$, p< 0.05, Fig. 9e) that exceeds the small but significant reduction in large phytoplankton production (-1.5 mmol C m$^{-3}$, p < 0.05) caused by decreased nutrients early in the year (nitrate and iron, Fig. 9c, d). The negative chlorophyll anomaly that starts in the spring (April) thus is due to the decreased large phytoplankton, which have a higher simulated Chl:C (0.027), offset by the increased small phytoplankton (Chl:C = 0.016) production anomaly. Because anomalies are insignificant later in the year for both size classes (late spring to fall), the spring signal dominates the seasonal cycle and results in a negative annually integrated chlorophyll anomaly (-0.09 mg m$^{-3}$, p < 0.05, Fig. 9a). This is consistent with the slightly negative annually integrated chlorophyll anomaly observed in satellite data (-0.02 mg m$^{-3}$, integrated green line) though those data exhibit a greater compensation between the large negative spring anomaly and a positive summer anomaly (green line in Fig 9a). Unlike in the NPTZ, the annual composite anomalies in the AG (Fig 6, red box) all fall within 1 $\sigma$ of the interannual variability except large phytoplankton production and the ratio of large to small phytoplankton production (Figure 7a). For these variables, this suggests that MHWs are not the largest source of interannual variability in the region. For example, small phytoplankton production has a weak composite anomaly and exhibits a wide range of variability during MHW years: a negative or near-zero anomaly during the years 1963, 1965, 1990, and 1991 and positive anomalies during the remaining events. Despite this variability, anomalies in large phytoplankton production are consistently more negative than anomalies in small phytoplankton production (with an exception in 1990), resulting in a decline of the large to small phytoplankton production ratio and a shift towards smaller phytoplankton similar to the NPTZ region. Indeed, in this region, the MHW impact is greatest on chlorophyll, large phytoplankton production and especially the ratio of large to small phytoplankton, which all differ significantly (p < 0.05) during MHWs compared to the mean state.

The density distribution of summer chlorophyll concentrations in the AG further supports the hypothesis of a shift in the phytoplankton assemblage toward small phytoplankton (Fig. 10a–b). For each phytoplankton size class there is one main mode (Large phytoplankton chlorophyll peak centered at 0.29 mg m$^{-3}$; small phytoplankton chlorophyll peak centered at 0.25 mg m$^{-3}$). The model suggests that climatologically, 42 % of the AG area has large phytoplankton chlorophyll concentrations > 0.4 mg m$^{-3}$, but that this proportion drops to 35 % during marine heat waves. This shift is associated with a significant reduction in the mean chlorophyll concentration of the large phytoplankton fraction in the region (-0.02 mg m$^{-3}$, $p < 0.05$, Fig. 10). In contrast, mean small phytoplankton chlorophyll concentrations in the AG remain virtually the same during marine heat waves. This result is supported by the observational OSP mixed-layer size-fractionated Chl $a$ measurements, which also displayed atypically low large phytoplankton (>5 µm) Chl during the 2015 warm blob (0.082 mg m$^{-3}$, star, Fig. 10a); in contrast, small phytoplankton Chl $a$ exhibits effectively no change (0.29 mg m$^{-3}$, star, Fig. 10b) during the same period compared to measurements from non-MHW years. These observed Chl values are lower than the simulated values and only sampled during the 2014–15 warm event. However, they are consistent with a transition of the phytoplankton assemblage toward the smaller size class in the AG region around OSP.

**4 Discussion & implications**

**4.1 Confinement of marine heat wave biological response to the transition zone**

Previous studies have demonstrated there is a decrease in primary production in the NPTZ caused by reduced nitrate concentrations during MHWs. During the "warm blob" atmospheric blocking by an atmospheric ridge (Le et al. 2019) decreased the wind-driven Ekman transport that generally carries nitrate from the northern AG southward, a process which otherwise supports up to 40 % of the new production (Ayers and Lozier 2010). Further, nitrate concentrations were reduced by warmer upper ocean conditions which drove a reduction in winter mixing (Amaya et al. 2021). Our results support these previous studies, with both observations (Line P, Argo floats) and the MOM6-COBALT model indicating lower nitrate concentrations during MHW across the AG (which also has lower iron) and the NPTZ (Fig. 6c). However, we show that chlorophyll and biological production anomalies are restricted to the NPTZ only (Fig. 6a,e,f). Our results suggest, that nitrate concentrations alone cannot explain the confinement of the biological anomalies to the NPTZ, and that the interplay between nitrate and iron limitation, more specifically the position of the nitrogen-to-iron limitation boundary (i.e., the boundary between the northern iron-limited regime and the southern nitrate-limited regime), controls the location of the strongest MHW ecosystem anomalies.

We find that production anomalies associated with MHW are strongest in the NPTZ because the influence of reduced winter nitrate supply is greatest in the region that seasonally transitions from iron limitation in early spring to nitrate limitation in summer. In the subtropical gyre south of the NPTZ, nitrate is depleted year-round so that nitrate concentration cannot decrease and impact biological production during MHWs. In the core of the AG, north of the NPTZ, annual production is iron-limited for large phytoplankton and mostly light-limited for small phytoplankton, thus changes in nitrate concentration have only a limited effect. In the NPTZ, however, nitrogen limitation starts earlier

and is more intense during MHWs, with nitrogen limitation factors that are about 20 % smaller during spring and summer of MHWs than in the climatology (Fig. 8h). As a result, the NPTZ is the region where primary production is most impacted by the decrease in nitrate associated with MHWs.

**4.2 Collapse of observation-based production misattributed to marine heat wave**

The northward expansion of the nitrate-depleted region during MHW introduces biases in float-based estimates of net community production (NCP) and export. Floats in the vicinity of the NPTZ nitrate front can easily sample both the high nitrate and nitrate-depleted regimes within a small spatial area (~300 km, Fig. 2) and over the course of a few

weeks or months. Float-based estimates of NCP (Appendix A) interpret nitrate changes sampled along the float trajectories as temporal changes, leading to a misattribution of this spatial variability in nitrate to seasonal biological drawdown. In Bif et al. (2019b), NCP was calculated using the winter to summer difference in nitrate concentration measured by 6 Argo floats in the vicinity of OSP and the NPTZ between 2008 and 2018 (see details on method in SI, Fig. 11). From these data, they concluded that there was a collapse in ecosystem production during the warm blob in

2015. However, this dataset only includes one float sampling the area in 2015 (float 5904125), and the trajectory of that float incidentally sampled the low nitrate biome in winter before shifting to the higher nitrate HNLC region in summer (brown track, Fig. 2). The sampling of these two distinct biomes is supported by the at-depth (>100 m) measurements of temperature, nitrate and salinity, which indicate that this float crossed into a new water mass in early 2015 (Fig. 3). As a result, the winter-to-summer change in nitrate along the float was artificially small and the NCP

calculation biased low in 2015.

We quantified the effect of the float shifting from the NPTZ to the AG on the NCP estimate by recomputing the NCP along the same float trajectories (latitude, longitude and time) but sampling three different nitrate fields to create "synthetic profiles". We thus obtain three other NCP estimates using nitrate from the climatological World Ocean

Atlas (black dashed), and from the model climatological (blue dashed) and interannual (solid blue) fields that can be compared to the observed Argo-based NCP estimate (red dashed, Fig 11). Note that for climatological profiles, only the month and day are used to sample the fields, while for the interannual field the year is also used. We find that most (>70 %) of the NCP reduction derived from these floats can be explained by sampling the climatological nitrate field, and that the apparent ecosystem collapse in 2015 is in fact a feature of the float trajectory that sample across the nitrate

north-south gradient (WOA vs. Argo). We performed a similar analysis using the MOM6-COBALT model, first sampling the model climatological nitrate field and found the same result: the sampling trajectory of the float leads to an artificial NCP collapse in 2015 compared to other years that did not sample across this gradient. When considering model interannual variability, and hence the effect of the marine heatwave, we find however an even stronger decline in NCP in 2015, suggesting that there is an NCP change caused by the "warm blob" itself. The model suggests,

however, that only 30 % of the decline in NCP can be attributed to the heat wave, while the remaining 70 % is attributable to the sampling across the NPTZ nitrate front.

**4.3 Shift in phytoplankton assemblage due to contrasting size-class response**

Our results suggest that during MHWs, there is a shift in the phytoplankton community toward the smaller size class across both biomes. Large phytoplankton primarily respond to changes in nitrate and iron limitation. In the NPTZ, large phytoplankton are greatly impacted, with a 13 % decrease in annual production (Fig. 6e, Fig. S4e) caused by stronger nutrient limitation during MHWs (iron from January–April, then nitrate from June–Dec, Fig. 8h). In contrast, small phytoplankton in both regimes respond to both reduced light limitation (in spring) and enhanced nutrient

limitation (in summer). Inside the NPTZ, where nitrogen limitation is strongest, small phytoplankton production increases in spring, but is inhibited through summer until the mixed layer deepens in fall, resulting in a modest 4 % decrease in annual small phytoplankton production. While the changes in the size-specific production anomalies are both within $1\sigma$ of the regional interannual variability there is a systematically greater decrease in large phytoplankton production that results in a large decrease in the ratio of large to small phytoplankton production (decrease of the order

of -1 $\sigma$, Fig. 7b).

In the AG, the annual anomaly in large phytoplankton production is small (-2 %), driven by low production in spring when the iron supply is decreased (Fig. 9e). This is likely because the reduced winds that were shown to reduce nitrate supply during MHWs (Whitney 2015; Le et al. 2019) also impacted the iron supply. Unlike nitrate, however, iron

returns to near-climatological levels by summer (Fig. 9d), which suggests that the sources of iron are decoupled from nitrate through the latter half of the year. As in the NPTZ, the spring small phytoplankton response is positive due to shallower mixed layers, however, in the nitrate-rich AG there is no summertime nutrient limitation of small phytoplankton and thus annual production is increased (+2 %, Fig. 6f, Fig. S5f). This contrasting response between the two size classes leads to a sizeable decrease in the large to small phytoplankton production ratio (decrease of the

order of -1.2 $\sigma$, Fig. 7a).

Across both regions, this shift implies that during MHWs, there is a reduced proportion of large phytoplankton such as diatoms and dinoflagellates versus greater proportions of smaller groups such as cyanobacteria and nanoflagellates. Evidence of this shift has been observed in the AG during the "warm blob" (Peña et al. 2019) which found higher

concentrations of cyanobacteria in the nitrate-depleted region of Line P. Further, the data presented in this paper show higher Chl $a$ concentrations in the smaller size classes at OSP (Sect. 3.3, 3.4). However, our work suggests this shift is more widespread, impacting both the AG and the NPTZ. Because diatoms and other large phytoplankton are known to support more productive food webs and more efficient biological carbon pumps (Boyd and Harrison 1999), their decrease would likely substantially affect the marine ecosystem structure and reduce carbon export potential. This in

turn, increases mortality risks for certain species, may promote geographical redistributions of fisheries, and can create challenging social and political environments stemming from the associated economic impacts (Frölicher and Laufkötter 2018). In the future, we should anticipate these ecosystem shifts as MHWs are expected to recur (Xu et al. 2021) and the atmospheric pressure systems associated with extreme events will increase in frequency (Giamalaki et al. 2021).


**Appendix A.**

Argo NCP Calculation

To calculate NCP, it is assumed that new production is fueled by nitrate supplied from the deep ocean during winter mixing (Dugdale and Goering 1967). Thus, the temporal change in nitrate stock from the winter, i.e. when the mixed

layer is deepest, to a given date must be due to net community production. This temporal change in nitrate is integrated over the top 75 m as follows:

$$NCP = r_{C:N} \int_0^{75} NO_{3_{winter}} - NO_3 \, dz \qquad (1)$$

Where $r_{C:N}$ is the Redfield ratio of carbon to nitrogen. This calculation assumes all changes in nitrate are due to NCP, ignoring potential lateral and vertical contributions from physical transport. An integration depth of 75 m is selected

to remain above the nitracline to limit the influence of transport, so that changes in nitrate above this depth can be largely attributed to biological processes. February was selected to be the winter month for each year as the mixed layer is often maximal between January and March; this simplification allows for a continuous time series to be calculated from February of each year. These choices are consistent with the previous study of Bif et al. (2019b).


**Author contribution**

AW and LR performed the model simulation and data analysis. AM provided OSP Chl *a* measurements. AW and LR prepared the manuscript with contributions from AM.


**Competing Interests**

The authors declare that they have no conflict of interest.

**Acknowledgements**

This study has been supported by the NASA ROSES 2018 EXPORTS award# 80NSSC17K0555 and 80NSSC17K0552, the National Science Foundation (NSF) project "Eddy Effects on the Biological Carbon Pump" award# 2023108, the Sloan Foundation, and the NSF Graduate Research Fellowship Program (NSF-GRFP). Funding that enabled AM participation on the Line P cruise in 2015 was provided by NSF award# OCE1334935. The authors thank David Luet and Enhui Liao for their help and technical support in developing and running the MOM6-COBALT

ocean simulation used in this study. The authors also thank the Geophysical Fluid Dynamical Laboratory and the MOM6 ocean modeling team for developing, maintaining and making the MOM6 ocean model publicly available on GitHub (https://github.com/NOAA-GFDL/MOM6-examples).

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

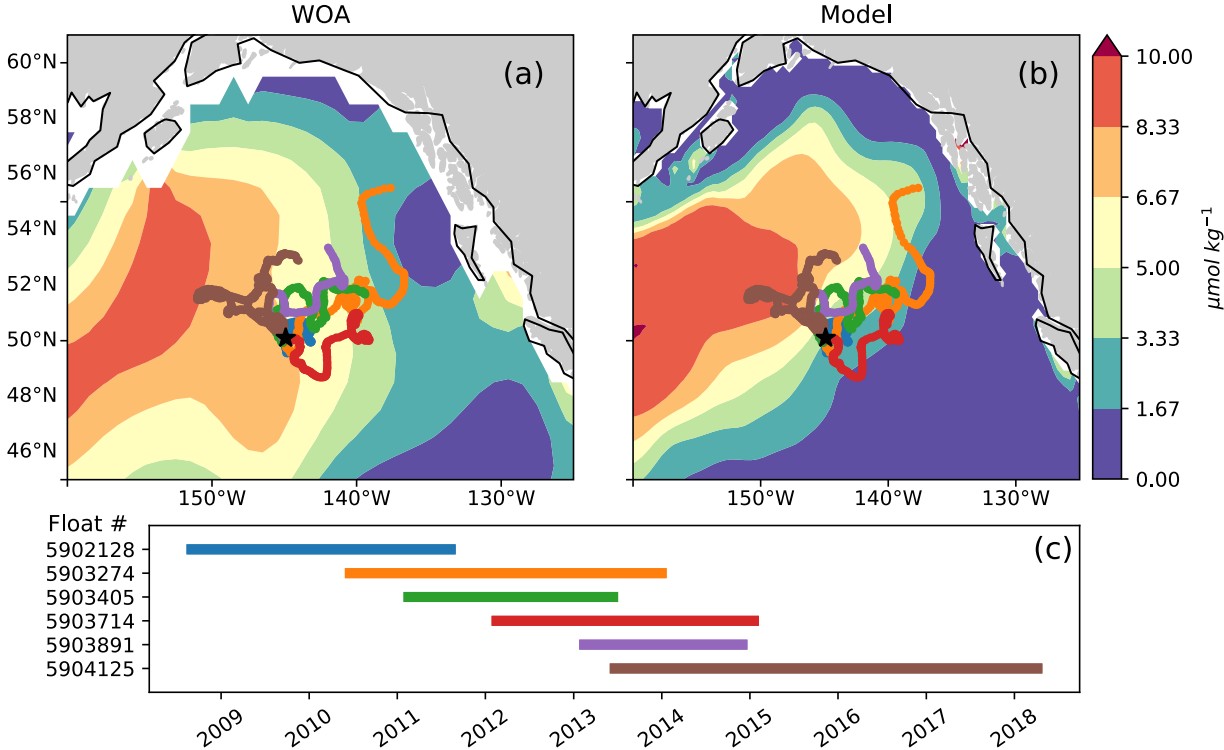

Fig 02. Argo float trajectories overlaid on late summer (August) surface nitrate concentrations in (a) the World Ocean Atlas, and (b) MOM6-COBALT. The sampling period of each float is shown in (c). Note that float 5904125 (brown) travels west across the east-west nitrate gradient, and is the only float sampling the region between 2015 and 2018.

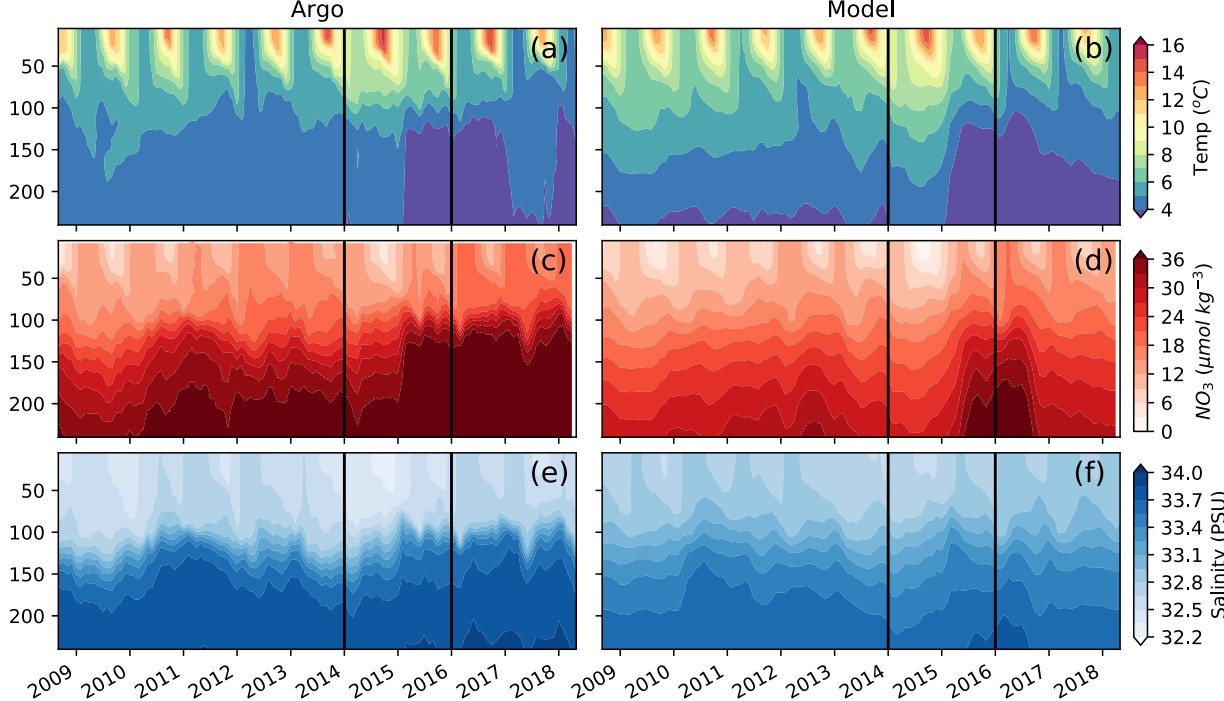


Fig 03. Comparison of observations from BGC-Argo floats shown in Fig 2 (left) and sampled along their trajectories in MOM6-COBALT (right) for temperature (a,b), nitrate concentration (c,d) and salinity (e,f). The "warm blob" period (January 2014 to December 2015) is delimited by vertical black lines.

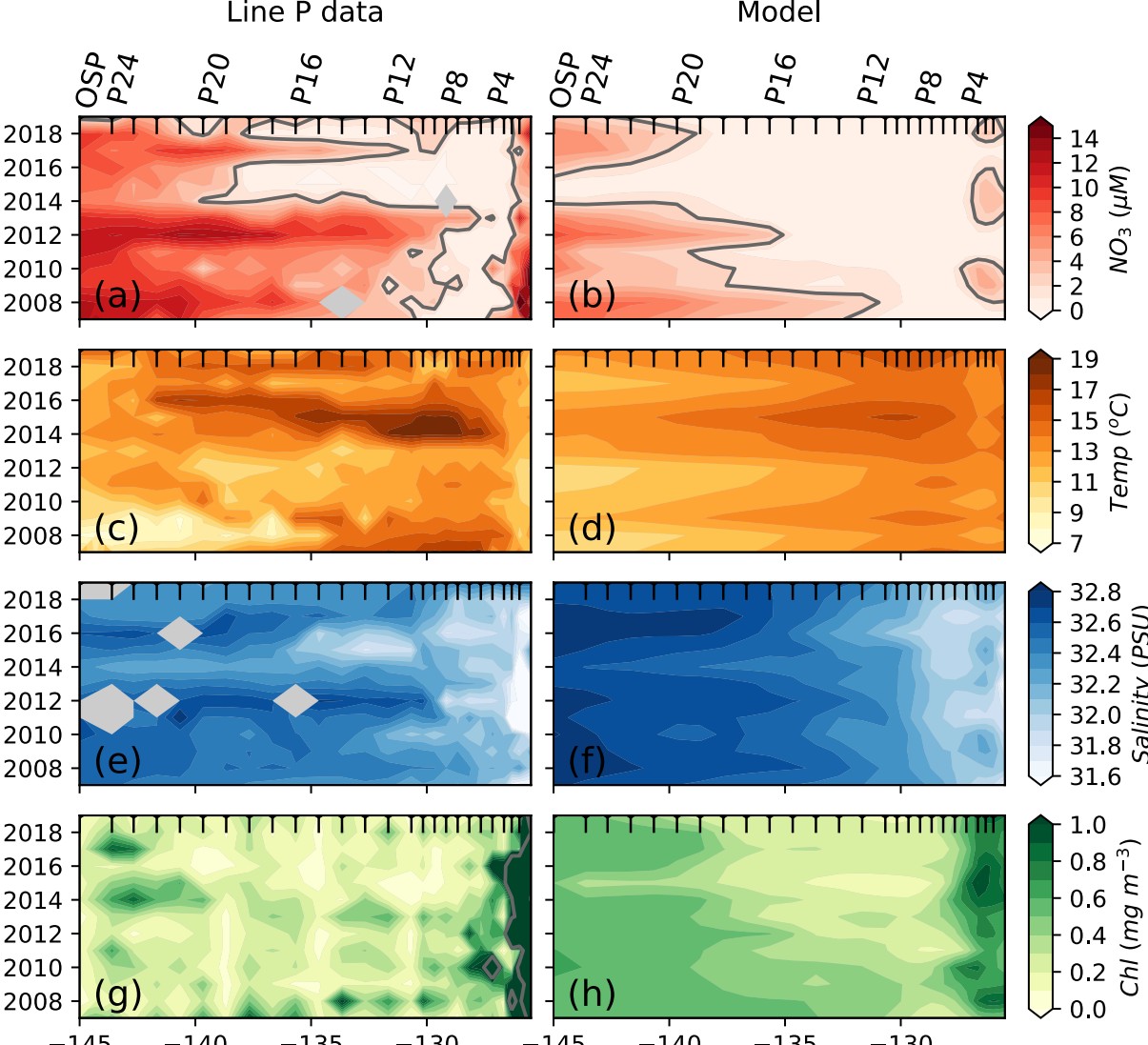

Fig 04. Impact of the 2014–2015 "warm blob" along Line P.  a) Surface nitrate concentration averaged across summer cruises (generally one in June, one in August) along Line P stations P1 to P26 (OSP). b) Same as panel a, but sampled in the model at the station locations and averaged from June through August. The 2 $\mu$M nitrate contour is shown as a solid gray line.  Other panels are the same as (a,b) for observed and modeled sea surface temperature (c,d), sea surface salinity (e,f) and  surface chlorophyll (g,h). Gray shading indicates lack of data. Black x-axis ticks indicate station positions. See Fig 1 for Line P station map.

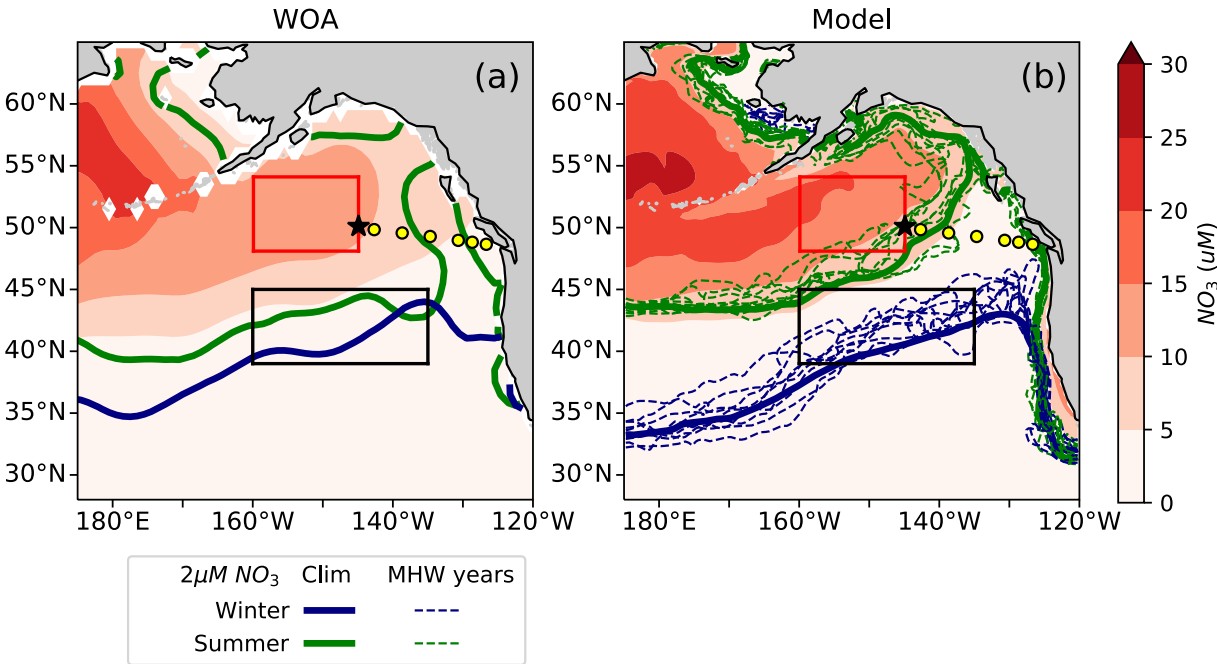

Fig 05. Annual mean surface nitrate concentrations in (a) world ocean atlas observations (WOA, Boyer, 2018) and (b) MOM6-COBALT model. The seasonal location of the 2 $\mu$M surface nitrate contour in February (blue) and September (green) are indicated for the climatology (solid lines), and all individual warm events (thin dashed lines). The nitrate-depleted region south of the 2 $\mu$M contour generally shifts further north in both winter and summer during warm events. The North Pacific Transition Zone (NPTZ, 39º−45ºN and 160º−135ºW) shown as a black box and the Alaskan Gyre (AG, 48º−54ºN and 160º−145ºW) as a red box. Line P stations and OSP are shown as described in Fig 1.

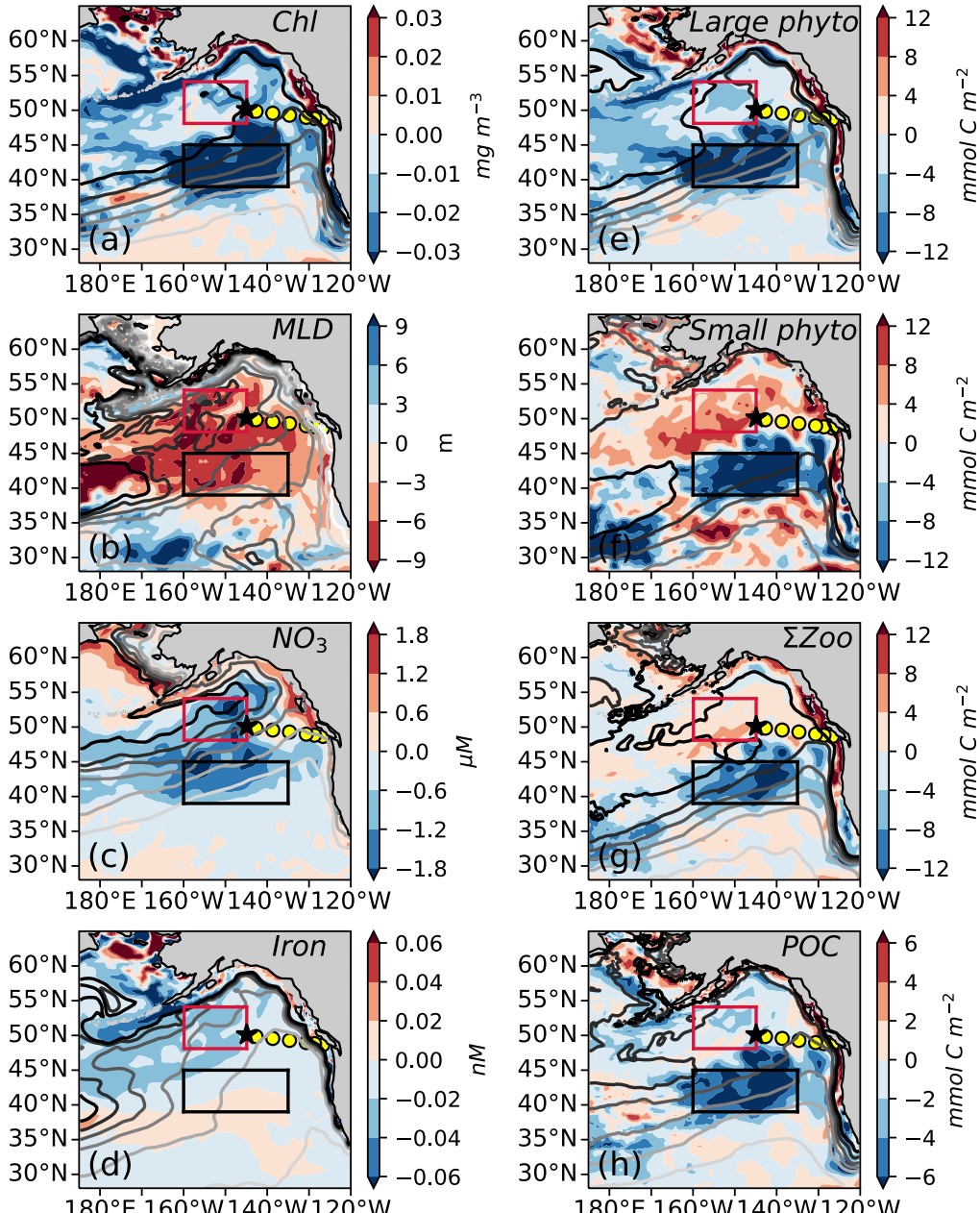

Fig 06. Modeled composite anomaly of the 9 marine heatwave (1958–2020) for (a) monthly surface chlorophyll concentration, (b) winter (Jan–Mar) mixed layer depth (MLD), (c) winter surface nitrate concentration, (d) winter surface iron concentration, (e) annual depth-integrated large phytoplankton production (0–100 m), (f) annual depth-integrated small phytoplankton production (0–100 m), (g) annual depth-integrated sum of large, medium and small zooplankton production (0–100 m), (h) annual particulate orgnic carbon (POC) export at 100 m depth. Each field is overlaid with contours of the mean climatological state with darker lines indicating higher values (see mean state maps in Fig S3). Line P, OSP as shown in Fig. 1; boxes for AG and NPTZ are shown as described in Fig. 5.

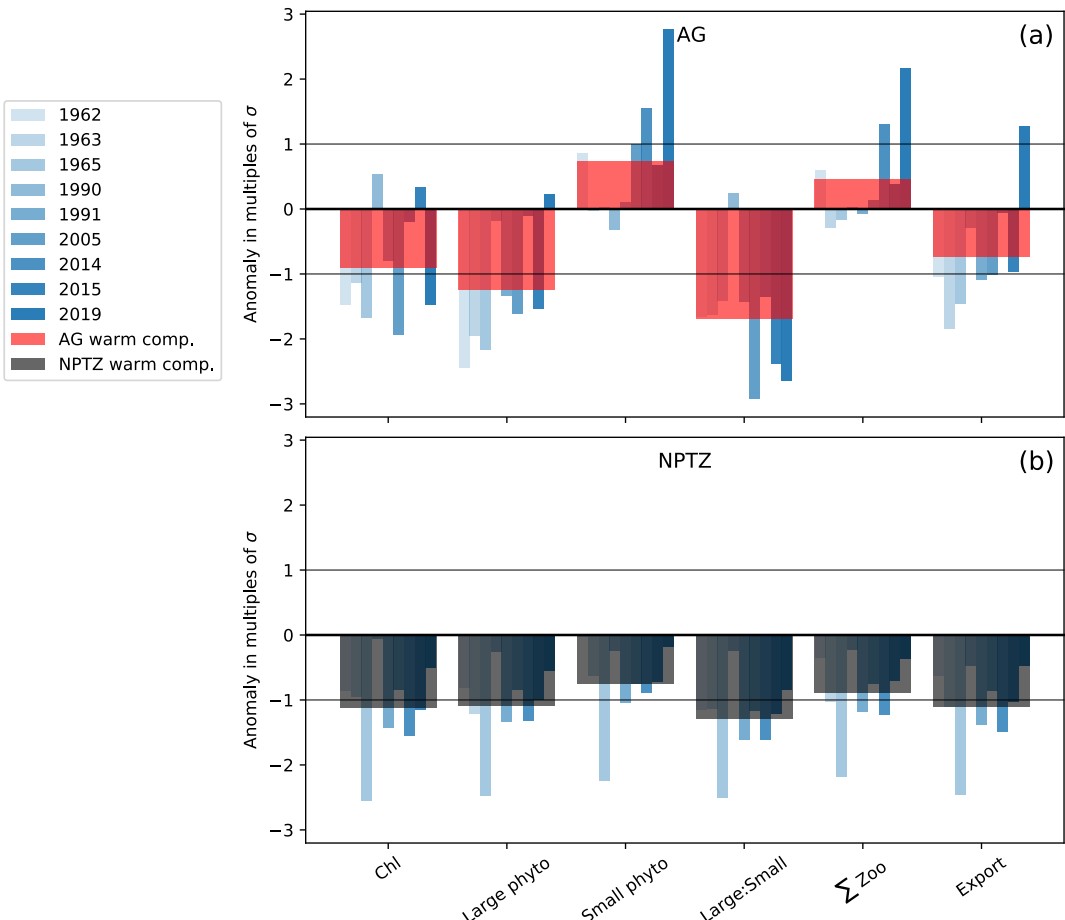


Fig 07. MHW anomalies for the 9 individual years (blue bars) and their composite (red and dark bars) in the (a) the Alaska Gyre (AG) and (b) the North Pacific Transition Zone (NPTZ). Anomalies are normalized by the regional interannual variability calculated as the standard deviation ($\sigma$) of the spatially averaged fields (i.e. values < -1 and >1 are anomalies that exceed $1\sigma$). Anomalies are shown for monthly Chl, as well as annually integrated large and small

phytoplankton production, large to small phytoplankton production ratio (Large:Small), total zooplankton production (Total Z prod) and export production.

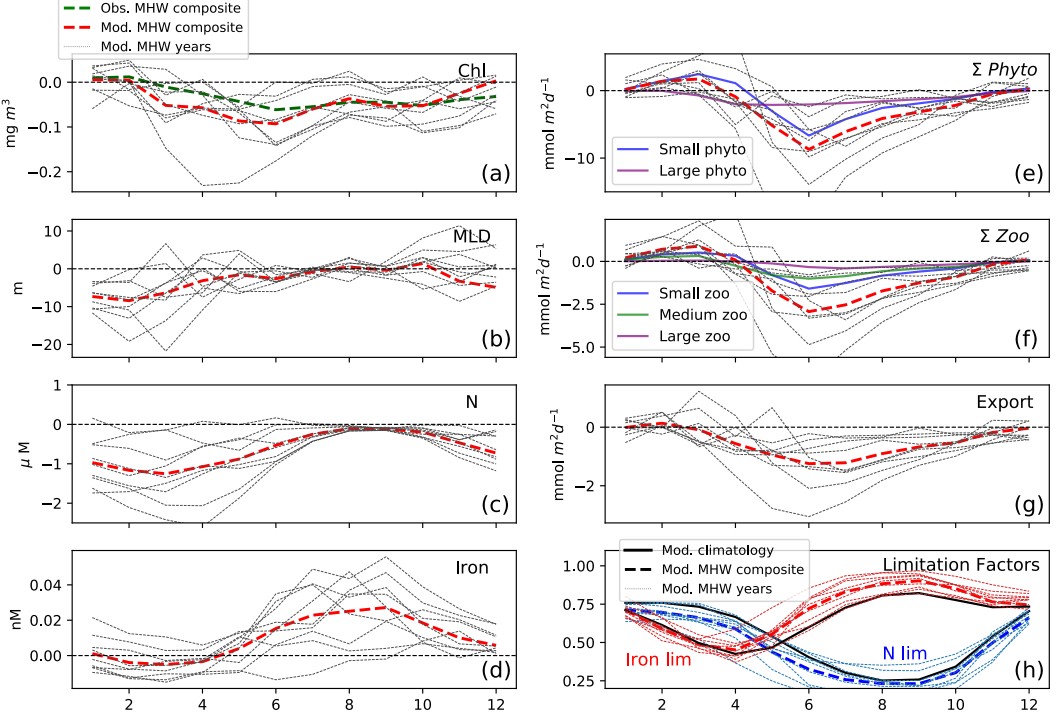

Fig 08. Seasonal response to MHWs in the NPTZ (black box shown in Fig. 5, 38°–48°N and 165°–135°W).
Composite anomalies of the 9-event MHWs for: (a) modeled surface chlorophyll (red) and observed surface chlorophyll (GlobColour, green),  (b) modeled mixed layer depth, (c) surface nitrate concentration, (d) surface iron concentration, (e) depth integrated phytoplankton production (0–100 m) with individual size classes (small in blue, large in purple), (f) depth integrated zooplankton production (0–100 m) with individual size classes (small in blue, medium in green, large in purple), (g) particulate export production at 100 m depth and (h) large phytoplankton nutrient limitation factors for iron (red) and nitrate (blue) for the mean climatological state (solid line) and the MHW composite (dashed lines). Across all panels, thin lines show anomalies for the 9 individual MHW years.

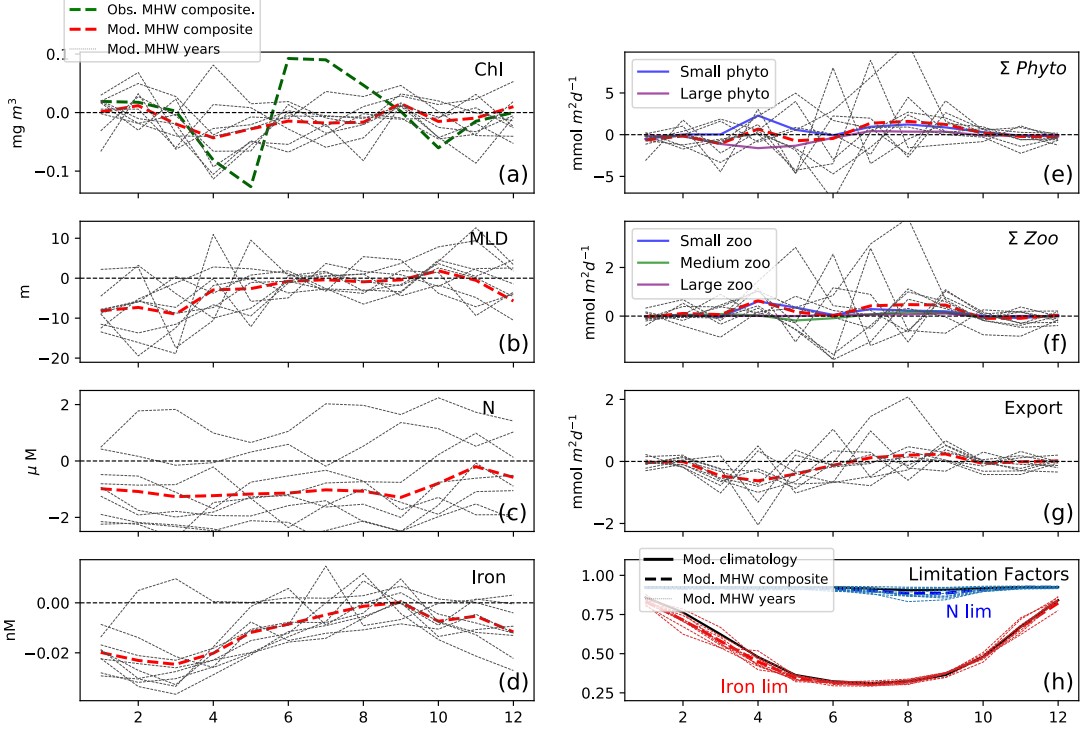

Fig 09. Response to marine heat waves in the AG (red box shown in Fig 5, 48°–54°N and 160°–145°W). Same as Fig. 8 for AG region.

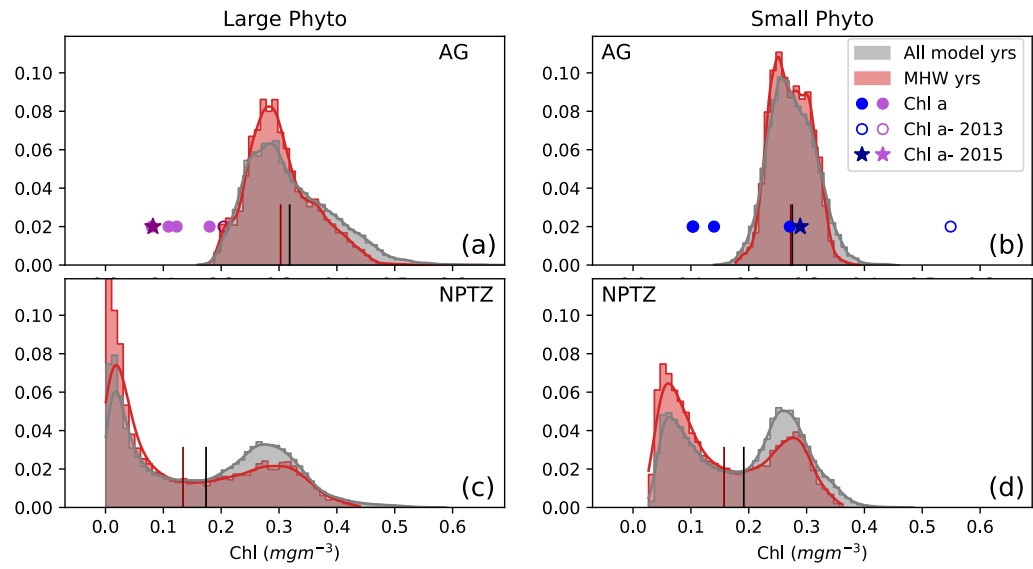


Fig 10. Observed and modeled summer (May–Aug) chlorophyll (mg m⁻³) contained in the large (left) and small (right) phytoplankton size fraction in two regions: (a,b) Alaska Gyre and (c,d) North Pacific Transition Zone (See

Fig. 5 for maps of zones). Model data are shown as normalized probability density functions for the MHW composite (red) and the climatology (gray). The mean of each is shown as a short vertical line on the x-axis (red, black respectively). Chl *a* observations from the six OSP cruises in the Alaska Gyre are shown as symbols on panels a–b at y=0.02 (data for the non-MHW years 2000, 2001, 2008, 2013, 2015, and 2018, are shown as filled circles while data from the anomalous 2015 warm blob and 2013 volcanic eruption are shown by a star and hollow circle respectively.


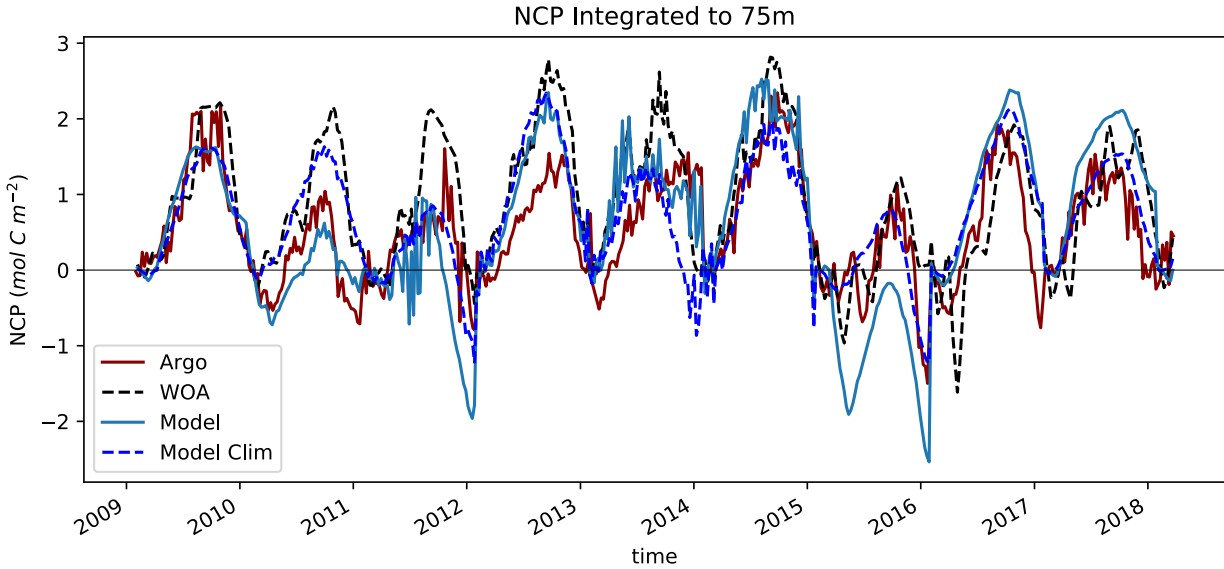


Fig 11. Net community production (NCP) calculated as nitrate drawdown from winter inventory (February) using BGC-Argo float data (dark red). These float trajectories were then used to sample the WOA climatological nitrate field (black dashed), the interannually variable MOM6-COBALT nitrate field (light blue) and the MOM6-COBALT climatological nitrate field (blue dashed) to get "synthetic profiles" with an apparent NCP calculated similarly.
