# Peer review of "Ecosystem impacts of marine heat waves in the Northeast Pacific"

_EGUsphere, 2022_

## Author Response (AR1)

Reviewer comments in blue

Author comments in black

We thank the editor and both reviewers for their time considering our manuscript "Ecosystem impacts of marine heatwaves in the Northeast Pacific." The reviews were generally supportive, and we believe that addressing the specific comments that were raised strengthened the paper. In particular we have added to and clarified the discussion of the significance of our results. We hope that the current version is acceptable for publication in Biogeosciences. Line by line responses to the reviewers are detailed below.

Reviewer 1 comments

1 This paper focuses on quantifying and explaining ecosystem impacts of marine heat waves in the NE Pacific, using a modelling approach supported by observational comparisons. The focus is on anomalies in chlorophyll and phytoplankton and their drivers. The topic is of high interest to the community. Several recent papers and theses look at the impacts of the 2014-15 Blob on productivity rates, ecosystem assemblages, and biogeochemical measurements like trace metals. This study broadens that work to consider a wider spatial area, assess drivers quantitatively, and provide some context for the results of previous work like decreasing nitrate observed by BGC-Argo floats. However, in many places in the paper, I was left wondering about whether the small changes detected were significant. Including statistical tests for significance would strengthen the paper, making the overall message more convincing. In general, the paper is well written, but I do have a few additional comments that I think could further improve it.

This paper reports many anomalies for marine heat waves based mostly on model results. Some of the anomalies are very small relative to the absolute concentrations or rates. Which ones are significantly different from zero? The paper would be strengthened by including a statistical test for significance in each case where a change associated with marine heat waves is reported, providing clarity for which changes are significant and which should be reported as no change within error. Examples include (but are not limited to):

• 2% lower large phytoplankton population in the AG

• 2% decrease in large:small phytoplankton in both regions

• 05 mg m-3 decline in chlorophyll in the NPTZ and 0.02 mg m-3 increase in chlorophyll in the AG – The AG value is especially small compared to mean chlorophyll in this location. Is the mean for MHW years significantly different from the mean for other years, given the fairly high variability in this region?

- Mean-MHW values throughout sections 3.3 and 3.4.With 9 MHW years and many non-MHW years in the model, it should be straightforward to calculate whether the MHW years are significantly different (perhaps a Mann-Whitney test or similar) in different months.

- 13% lower large phytoplankton production

- The location of the 2 uM nitrate contour in Figure 5b.Given the variability in the location of this contour in the 9 MHW events, is the mean significantly different from the all-year average?

We address the significance of the listed statistics in the new proposed Figure 7 which compares the size of the marine heat wave anomalies to the interannual variability in each region. This figure shows that chlorophyll and phytoplankton production anomalies tied to marine heatwaves can for some of these events reach relatively high values compared to the interannual variability exhibited in each region (i.e. of the order of 1 to 2 standard deviations). For instance, in year 1965 there are strong negative production anomalies (>2.2 σ for large phytoplankton production; >2.1 σ for small phytoplankton production) in the NPTZ. Yet, on average across the 9 events, and in contrast to prior work using satellite-based chlorophyll (e.g. Whitney 2015), we find that chlorophyll, phytoplankton and zooplankton production respond relatively modestly to marine heatwaves in both regions (variability of the order of 1 standard deviation or lower, new Figure). Notably, however, we find a relatively robust decrease in the ratio of large phytoplankton to small phytoplankton production across all events and in both regions (meets or exceed 1 standard deviation), suggesting that marine heatwaves in the northeast Pacific result in a shift of the phytoplankton assemblage favoring small phytoplankton production.

Note on method for the new figure: To calculate the composite anomalies for the 6 ecosystem variables, we detrended the model time series, calculated at each model grid point. We then used these detrended data to update figures 8, 9 & 10, which also examined composite anomalies. The area averaged composite anomaly was compared to the area averaged standard deviation of the monthly model output (1958 - 2020), which we used as a metric of interannual variability.

In addition to the addition of Figure 7, the following lines were changed to reflect the inclusion of the new figure:

Boundaries of nitrate and iron limitation are discussed, but I'm unsure of how these are defined. Nitrate limitation may be defined by the 2 uM contour line, though this should be explicitly stated. I'm not sure what iron value would be considered limiting here. In general, the iron concentrations in the model (Figure S2) seem very high with modelled values around 100-150 nM iron in winter, whereas typical values for dissolved Fe in the region appear to be < 1 nM (see https://doi.org/10.1016/j.marchem.2015.04.004 for example). Is the model iron limiting anywhere? I could not discern the hatching in Fig. 6 discussed in Line 222 or the gray and purple lines discussed in Lines 231-236.

We apologize for the confusion here which was also mentioned by Reviewer 2. This paragraph references a figure showing limitation boundaries that was removed prior to the original submission. This paragraph has been edited to reference the Line P data analysis only. Lines referenced above were removed. The new paragraph is located at L327-333.

How are the limitation factors shown in Figure 7h defined?  This section, along with that around (former Line 58), also caused me to wonder about the role of light limitation.  Is the NPTZ actually iron limited before the spring bloom or is it light limited then?  Is there grazing limitation that is important to controlling the size of the spring bloom in the NPTZ?

The nutrient limitation factors are computed in the model using kinetics relationship linking growth rate to nutrients. For instance, nitrate limitation is based on a Michaelis Menten formulation. We have updated the methods section 2.3 L113-118 to clarify how they are computed.

> Phytoplankton growth is explicitly modeled as size-dependent functions of light, temperature and nutrient limitations (nitrate, ammonia, phosphate, etc.). Small phytoplankton are simulated to be efficient nutrient and light harvesters (Munk and Riley 1952; Geider et al. 1997) in contrast to large phytoplankton, which are parameterized to grow quickly in response to abundant nutrients. Notably, in the study regions, this results in large phytoplankton being sometimes iron limited while small phytoplankton are not. The limitation factors are output from the model as a number between zero and one, with zero indicating complete limitation, i.e. no phytoplankton growth.

Yes, you are right, iron is the main limiting nutrient in spring in the NPTZ, as can be seen in figure 7. However, as we state in section 3.3 light limitation is indeed a significant factor before the spring bloom that we have chosen to discuss in terms of mixed layer depth, (shallower during marine heatwaves) as follows (L273-277) :

> The model suggests that marine heat waves promote the growth of small phytoplankton and small to medium-sized zooplankton in early spring before dropping in summer–fall (Fig. 8e,f), due to the shallower mixed layer in winter and early spring (-10 m, Fig. 8b) that relieves light limitation and spurs small phytoplankton production (a positive production anomaly of +2 mmol m$^{-2}$ d$^{-1}$, Fig. 8e).

Minor suggestions:

Line 92:  I was very surprised to read that chlorophyll data was not available for 2008-2010 from Line P.  I contacted chief scientist, Marie Robert, to ask.  She looked into it and has found that the data exists but that there is a problem with some of the summary .csv files.  Some individual casts seem to contain the data but the whole cruise files do not.  She is working on updating the files.  I suggest you contact her directly for updates: Marie.Robert@dfo-mpo.gc.ca

We have reached out to Maire Robert and Figure 4g is updated with the new data, which aligns with the model and reinforce our initial conclusions.

Section 2.5: Suggest BGC-Argo rather than bioArgo.  Suggest referencing Appendix A here.

Changed as suggested throughout. Reference to Appendix A added.

[Former] Line 175: Fig. X

L190 Corrected to Fig. 4b

[Former] Lines 203-214: Suggest mentioning in this section that the high nutrient regime near the coast is temporally variable and mainly controlled by the timing of upwelling events.

L218-219 changed to:

> The Line P program's June and August cruises sample three regimes (Fig. 4): the temporally variable, high-nitrate near-shore region (>10 $\mu$M at ~125° W)...

Figure 1: I'm unclear why the bounding box for the average anomalies shown in panels a and b is different from either box shown on the maps. The targeted area appears to be mainly in the NPTZ. I suggest that it would be more illuminating to show the time series for the black NPTZ box in panels a and b rather than a different region that is not used in further analysis. Panel b: The region for this anomaly is probably the same as for panel a, but it would be good to state that. I think the location of OSP should be 50oN 145oW, not 50.1oN 149.9oW.

The authors agree that it was confusing to include the NPTZ and AG boxes in Fig 1. Descriptions of these regions have been moved to Fig 5. We have also removed the time series of chlorophyll anomalies (panel b) to simplify the text. Figure 1 is now clarified to show the time series from which marine heat waves were selected using the box 35° to 46° N and 150° to 135° W. This specifically follows the selection and analysis of Xu et al. as detailed in the methods. The location of OSP was corrected as suggested.

Figure 2: the colours of the float trajectories in the upper panels should match those in the lower time period panel. In particular, the brown colour in the upper panels is orange in the lower panel. Colour bar label is cutoff for panel b.

Yes, thank you. This was a mistake and colors have been corrected to match and we have replaced the labels to ensure they are fully visible.

Figure 3: Suggest adding property labels to the colour bars, i.e. not just units.

We added labels (PSU, NO3 and Temp) to colorbar labels

Also for the y-axis of Figure 10.

We added "NCP" to Figure 10 y axis

Figure 4: Colour bar labels are cut-off. Figure 5: Colour bar labels and legend are cut-off.

This has been corrected to make sure full labels are visible on all figures.

Reviewer 2 Comments:

I think this is a good paper that is publishable with more or less minor revisions. Some aspects of the methodology are insufficiently explained. The terminology is confusing in some places, and some unnecessary jargon is used (see details below).

Major points:

R1-1) There are some important details missing from the description of the methods and the data. Most importantly, a marine heatwave is defined as "anomalies that exceed 1 standard deviation for 5 months or more". But standard deviation of what and anomalies relative to what? The obvious answer is relative to a climatology calculated over the period of the ERSSTv4 data product, but that needs to be stated explicitly, and which years of this data product were used does not appear to be stated anywhere.

And is there an area threshold? Is the criterion applied point-by-point, or only to the regional mean? Would it be a heat wave if only 1 grid point exceeded the threshold?

Thank you. The reviewer is correct that we reference the climatology of the ERSST product when calculating the standard deviation and anomalies. We have clarified that we use an area mean SST anomaly, so that a single grid point exceeding the 1-standard deviation for 5 months threshold would not qualify in our study as a MHW. We also clarify that this anomaly is relative to the climatology of the specified region from 1958-2020. Section 2.1 now reads.

L78-82:

> Following the method of Xu et al. 2021, we calculate the area mean sea surface temperature anomalies (SSTa) relative to the climatology of the region $35^{\circ}$ to $46^{\circ}$ N, $150^{\circ}$ to $135^{\circ}$ W using the monthly data from 1958 – 2020 of the Extended Reconstruction SST dataset (ERSSTv4, Huang et al., 2015). Northeast Pacific marine heatwaves are defined as periods when the monthly deviation relative to the climatology exceeds 1 standard deviation for 5 months or more. The same method is used to detect marine heatwaves in the ocean model (see Sect. 2.3 for model details).

(And why does the "MOM6-COBALT climatology" in Figure 10 appear to have interannual variability?)

Figure 10 depicts net community production (NCP) calculations derived from the float trajectories (lat, lon, and time) as sampled in the world ocean atlas (WOA) and the model climatological nitrate field vs the interannually varying model field. In the case of the WOA and model climatology, only the month and day were used to sample the field, while in the interannually variable model field, the year was also used. The fact that this figure appears to show interannual variability while using climatological nitrate fields highlights the observation bias tied to the float sampling trajectories. We conclude that 70% of the NCP collapse calculated

during MHWs is actually attributable to the sampling bias and not interannual variability. We've updated the caption to read:

> Fig 10. Observed and modeled summer (May–Aug) Chl (mg m-3) contained in the large (left) and small (right) phytoplankton size fraction in two regions: (a,b) Alaska Gyre and (c,d) North Pacific Transition Zone. Model data are shown as normalized probability density functions for the MHW composite (red) and the climatology (gray). Chl a observations from the six OSP cruises in the Alaska Gyre are shown as symbols on panels a–b at y=0.02 (data for the non-MHW years 2000, 2001, 2008, 2013, 2015, and 2018, are shown as filled circles while data from the anomalous 2015 warm blob and 2013 volcanic eruption are shown by a star and hollow circle respectively.

And we have clarified this at the end of section 4.2, L396-410.

"The model was spun-up using three repetitions of ... 1958 to 1985" (112). But spun up from what? From rest? 81 years doesn't seem very long to spin up a global ocean model. And why go to the trouble of initializing short-lived (i.e., insensitive to initial conditions) biological tracers from an ESM piControl (116-117), but not the physical ocean? I think it would make more sense to use the ESM piControl data to initialize the physical ocean, or 1958 of the historical run.

We use WOA to initialize temperature and salinity for the physical dynamics as this run uses realistic historical atmospheric forcing (Jra) and we want to use a realistic initial ocean. Using ESM piControl for the physical initialization would not necessarily allow us to have realistic T&S fields matching observations. We have clarified this in section 2.3.

 L112-119:

I find it difficult to believe that there are no Line P chlorophyll data before 2011 (175-176). Line P is one of the longest-running ocean time series programs, and the basic methodology for chlorophyll concentration has not changed in half a century.

As pointed out by reviewer #1, the files published by the Line P community were incomplete. We have reached out to Marie Robert and she will send us the updated files Figure 4g includes these updated data in the revised manuscript. Preliminary analysis of that data is consistent with the model results and confirm our conclusions.

Satellite chlorophyll data should be available back to 1996 or 1997. "GlobColour" is referred to several times in the figure captions but never in the main text.

This oversight was corrected and a description of the data product was added to section 2.5,

L157-159.

> Satellite chlorophyll observations (1997-2020) are from the GlobColour dataset (http://globcolour.info) which has been developed, validated, and distributed by ACRI-ST, France (Maritorena et al. 2010).

Are these really all of the Argo floats available in this region? Or is there some other selection criterion being applied that is not spelled out here (e.g., availability of nitrate data or data within a certain area)? I find it hard to believe that these are the only Argo floats deployed in this region over an 11 year period.

While there are other CORE-Argo floats that have sampled the region, the selected floats are the only available BGC-Argo floats that also sampled nitrate. We have updated section 2.5, to specify that we use BGC floats with the nitrate sensors.

L152:

This study makes use of the 2008–2018 series of BGC-Argo floats with nitrate sensors deployed near OSP (e.g., Fig. 9).

R2-2) The Abstract ends by saying that "primary production anomalies modify the allometric phytoplankton distribution, resulting in a 2 % decrease in the ratio of large to small phytoplankton in both regions". Firstly, this seems like a very small change to emphasize as a key point in the Abstract: I am wondering if it is a mistake and it should be 2X or 20%. Secondly, it isn't easy to tell whether this passage is talking about production or biomass, and seems to shift arbitrarily between the two. Finally, where exactly in the main text is this assertion substantiated? Figures 7 and 8 illustrate the seasonal decoupling of large and small phytoplankton production, but can not be used to directly infer the Large/Small ratio of either biomass or production. Figure 9 shows only summer data.

To address the significance of our results, as was also asked by Reviewer #1, we have proposed a new figure 7 which compares the size of the marine heat wave composite biological anomalies to the interannual variability of each region. This figure shows that chlorophyll and phytoplankton production anomalies tied to marine heatwaves can for some of these events reach relatively high values compared to the interannual variability exhibited in each region (i.e. of the order of 1 to 2 standard deviations). For instance, in year 1965 there are strong negative production anomalies (>2.2 σ for large phytoplankton production; >2.1 σ for small phytoplankton production) in the NPTZ. Yet, on average across the 9 events, and in contrast to prior work using satellite-based chlorophyll (e.g. Whitney 2015), we find that chlorophyll, phytoplankton and zooplankton production respond relatively modestly to marine heatwaves in both regions (variability of the order of 1 standard deviation or lower, new Figure). Notably, however, we find a relatively robust decrease in the ratio of large phytoplankton to small phytoplankton production across all events and in both regions (meets or exceed 1 standard deviation), suggesting that marine heatwaves in the northeast Pacific result in a shift of the phytoplankton assemblage favoring small phytoplankton production

See the response to Reviewer 1 for line by line details on where this figure is referenced in the text.

(BTW "allometric phytoplankton distribution" here is a good example of unnecessary jargon: "phytoplankton size distribution" would suffice. And if one wishes to get dogmatic, the anomalies do not "modify" the size distribution. This sort of quasi-teleological confusion of subject and object is characteristic of inexperienced authors receiving inadequate guidance

This has been corrected.

L24:

These primary production anomalies modify the **phytoplankton size distribution**…

(see also 208, "Salinity maintains ...")).

L223 This has been corrected "salinity maintains a lateral gradient" to:

There is a lateral gradient in salinity across the region,

R2-3) The interaction of the N and Fe cycles is sometime characterized in superficial terms, although I think the overall conclusions are mostly sound. It might help to spend a few sentences in the Introduction sketching out a conceptual model of how the authors think the overall system works.

We address these dynamics in the introduction when we introduce the two study regions. In particular we discuss the transition that occurs seasonally in the NPTZ from spring, when nitrate is abundant, to summer, when nitrate is depleted.

L56-62

The AG is a high nutrient, low Chl (HNLC) region, characterized by high nitrate concentrations, but moderate primary production throughout the year due to iron limitation that prevents the development of a strong spring bloom (Martin and Fitzwater 1988; Harrison 2002; Boyd et al. 2004, Peña and Varela 2007). In contrast, the NPTZ is a region characterized by strong seasonality in nitrate and Chl due to the seasonal biological consumption and the Ekman transport of nutrients (Polovina et al. 2008, Chai et al. 2003; Ayers and Lozier 2010). As a result, the NPTZ evolves from a subpolar-like, iron-limited biome when nitrate is abundant in spring to a nitrate-depleted, subtropical-like biome in summer,

On [Former] L164-165, would not a prolonged period of stratification also result in depletion of surface iron concentrations? In the absence of significant aeolian sources I think it would. However, it would also tend to drive the system towards N limitation even in the absence of new aeolian Fe.

While we note that we do have climatological aeolian iron deposition that affects the iron limitation, it is true that prolonged stratification would result in depletion of surface iron as well. We have changed the wording to the more general "nutrient" instead of "nitrate".

L179

…which posited reduced surface **nutrient** concentrations as a driver of reduced primary production and Chl concentrations during the "warm blob".

It also seems to be implied that only large phytoplankton are subject to iron limitation (130-135), which I think is questionable. Iron is potentially limiting for nanophytoplankton even if iron limitation is the main driver of the dominance of diatoms or nanophytoplankton. On 268 it is

stated that "small phytoplankton are not simulated with iron limitation" so possibly the lack of Fe limitation is by construction in this model. If this is the case it should be stated up front in the Methods.

In this model, the iron deficiency for each size class of phytoplankton is calculated explicitly, however in the study regions, iron is never limiting for small phytoplankton.

We have updated section 2.3 to state this clearly.

L113-121

> Phytoplankton growth is explicitly modeled as size-dependent functions of light, temperature and nutrient limitations (nitrate, ammonia, phosphate, etc.). Small phytoplankton are simulated to be efficient nutrient and light harvesters (Munk and Riley 1952; Geider et al. 1997) in contrast to large phytoplankton, which are parameterized to grow quickly in response to abundant nutrients. Notably, in the study regions, this results in large phytoplankton being sometimes iron limited while small phytoplankton are not. The limitation factors are output from the model as a number between zero and one, with zero indicating complete limitation, i.e. no phytoplankton growth. There are also three zooplankton size classes of which large (>2000 $\mu$m) and medium (200 to 2000 $\mu$m) make up the mesozooplankton pool with a third, separate small zooplankton class (<200 $\mu$m) all of which consume phytoplankton using size-related predator-prey relationships.

The model does include iron limitation for nanophytoplankton, which are simulated as diazotrophs, however, we do not discuss impacts of marine heat waves on that size class in this study as they are a small proportion of total primary production in the study regions.

The limitation factors are never really explained. I assume this means a number between 0 and 1 where 1 means N or Fe replete and 0 means no growth, but this should be clearly stated in the Methods. (On a terminological note, I think "nitrate limited" and "nitrate limitation" should be changed to "nitrogen" across the board.)

We have clarified the limitation factors in section 2.3 (see last point). However, we agree that we should change our analysis and discussion to "nitrogen limitation" instead of "nitrate limitation" which does not include the impact of ammonia. Changing to the more general nitrogen limitation slightly changes Fig 7h and Fig 8h, however, the result remains the same: In the NPTZ (Fig 7) iron is limiting in the spring only while in AG (Fig 8) iron is always limiting.

In the last paragraph of section 3.2, the terminology is sometimes vague or confusing, wrt what is meant by a "boundary". On 223, the "2 uM nitrate boundary" could be "2 uM nitrate contour". In the next sentence, "nitrate boundary" occurs without any context. I assume this means the boundary between regions of N and Fe limitation, but it could be spelled out more clearly. This is an example of a place where adding a few more words could increase clarity substantially. The last few sentences (226-229) read like a description of the model solution, and this seems like a missed opportunity to state what the authors think is happening in terms of physical processes (see also 339-343).

We apologize for the confusion here, which Reviewer 1 also pointed out. This paragraph references a figure showing limitation boundaries that was removed prior to the original submission. This paragraph will be cut to reference the Line P data analysis only.

L321-336:

> The Line P data support the model result and show an expansion of the nitrate-depleted region during the 2014–2015 "warm blob" (Fig. 4), leading to a westward shift of the 2 $\mu$M boundary to 140º W in 2014 (vs a location of ~130º W in the other years). In the model, this westward shift of the nitrate boundary is overestimated, extending past 140º W. Thus, in both the observations and model this implies that nitrate becomes depleted inside the climatological boundary of the HNLC AG. The HNLC region can therefore be considered to contract while the nitrate-depleted region expands.

Some details:

10 and elsewhere I would change "Alaskan gyre" to "Alaska gyre" across the board

Changed

15 change "limitations" to "limitation"

L15: Corrected to limitation

[Former] 17 delete "climatologically" or change it to e.g., "usually" or "chronically"

L18: Corrected to "already"

18 "Contrastingly, we find that ..." conversely? in contrast? by contrast?

L18: Corrected to "in contrast"

19 maybe change "lower light limitation" to "higher mean irradiance"

L19: Corrected "lower light limitation" to "weaker light limitation"

[Former] 20 change "allometric phytoplankton distribution" to "phytoplankton size distribution"

L24: Changed to phytoplankton size distribution

[Former] 26 not sure "recorded" is the appropriate term here; how many of these were recognized as such when they occurred?

L30: Changed to: on record

31 " a redistribution of marine biogeography " ???

L35: Changed to: a shift in marine species geographical distribution

32 delete "geographical"

Deleted

35, 37 "Chlrophyll"

All instances of "chlorophyll" are corrected to "Chl" in agreement with rest of paper

36 change "demarks" to "demarcates"

L39 Changed to demarcates

37 delete "Pacific"

Deleted

[Former] L38 change "nitrate surface concentrations" to "surface nitrate concentrations"

L41 Changed to surface nitrate

[Former] L48-50 this sentence is very awkwardly worded

L49 -51 Changed to:

> This bottom-up explanation does not explain why the decrease in Chl was highly localized (confined to the NPTZ) while anomalously low nitrate concentrations extended 600 km north (into the AG) of any significant Chl anomalies (Peña et al. 2019).

[Former] L57 change "Ekman-driven transport" to "Ekman transport" or "Ekman flow driven transport"

L60 Changed to Ekman transport

60-61 I would consider also citing Glover et al 1994 (10.1029/93JC02144) here (Bograd et al appears to be missing from the ref list)

Glover citation has been added and Bograd citation added to reference list

[Former] 67, 387 change "contrasted" to "contrasting"

L70 & L416 Changed to contrasting

127 delete "re-"

Corrected

132-133 delete "and are efficient ... Geider et al., 1997)"

This is significant to the resulting response between the two size-classes. We've moved this detail, however, to the biogeochemical model description in section 2.3 where it's better suited.

[Former] 158, 160 mmol kg^-1 should be umol

L174 & 175 Changed. Figure 3 units also corrected to umol

159 add a ' on "floats"

Corrected

171-172 "nitrate concentrations are near-zero for most stations (P4–P20)" Is this unusual? Don't some of these stations always see drawdown in summer? (e.g., Pena and Varela 2007).

Figure 4a shows the stations (P4-P8) that usually exhibit depleted nitrate during the summer cruises, while during the MHW, the nitrate depletion extends to P20. We've added a reference to Figure 4a.

[Former] 186 add "North" before "American"

L199: "North" Added

[Former] 191 change "biophysical" to "biogeochemical"

L206 Changed to biogeochemical

205 change "values" to "concentrations"

L220 Changed to concentrations

207 "> 5 mg m^-3" Is this a mistake? This is an extremely high concentration for an open-ocean environment.

L222: Yes. Typo was corrected to 0.5 mg m-3

211-214 this assertion seems disconnected from the preceding text; not clear what its relevance is

Here we wish to make it clear that while there are similar spatial patterns of low/high nitrate regions, the model exhibits a bias towards lower nitrate concentrations across this region in comparison to the observations. This text has been corrected to the following:

L226-229:

> However, we note that the modeled surface nitrate concentration is generally lower in comparison to the Line P data, with maximum values rarely exceeding $8\mu M$ versus $15\mu M$ in the observations (Fig. 3a–b) consistent with the annual mean nitrate bias mentioned above.

[Former] L217 add a "~" before "130 W"?

L232 "~" Added

[Former] L219 "nitrate becomes more depleted" more than what? (unclear antecedent)

This was corrected to the following:

L234-235:

> However, in both the observations and model this implies that nitrate becomes depleted inside the climatological boundary of the HNLC Alaskan Gyre.

[Former] 250 not clear what is meant by "in this region of the model"

L264: Removed "of the model" as this section is entirely about the AG

[Former] 267 "the limitation factor is significantly lower (-0.06)" significant by what criterion? P<what?

L281 Changed to:

> and the limitation factor is .06 significantly lower (~ 1 standard deviation).

[Former] L285-287 "Lg Chl" and "Sm Chl" appear only in this one place,

L333: Corrected to be "large phytoplankton Chl" and "small phytoplankton Chl" respectively.

as does "chl" (elsewhere Chl)

Corrected

[Former] 286-296 "southern-like" and "northern-like" appear only in this paragraph and are not defined or explained

L311: southern-like was changed to "subtropical-like" and northern-like to "subpolar-like"

[Former] 306 specify mmol of C or N

L321: Corrected to mmol C

[Former] 311-313 another very awkwardly worded sentence

L326-328: Changed to:

> This is consistent with the slightly negative annually integrated Chl anomaly observed in satellite data (-0.02 mg m$^{-3}$, integrated green line) though those data exhibit a greater compensation between a large negative spring anomaly and a positive summer anomaly.

[Former] 333-336 Does this sentence make sense? It reads like it is sort of arbitrarily combining different levels of causation. If there is a clear hypothesis as to "A leads to B leads to C", it would be better to express it that way.

This sentence combined too many ideas and has been broken into two parts.

L356-357:

During the "warm blob" atmospheric blocking by an atmospheric ridge (Le et al. 2019) decreased the wind-driven Ekman transport that carries nitrate from the northern AG southward that normally supports up to 40 % of new production (Ayers and Lozier 2010). Further, nitrate concentrations were reduced by warmer upper ocean conditions which drove a reduction in winter mixing (Amaya et al. 2021).

[Former] 350 add "concentration" after "nitrate"

L373: Added

[Former] 360 "changes sampled along the floats" along the floats' trajectories?

L384-385 Added "trajectories"

[Former] 364 change "this data" to "these data"

L388: Corrected

398 comma in wrong place

L426 Corrected

[Former] 416-417 I'm not sure this sort of editorializing is necessary, and I doubt that it is discussed by Frölicher and Laufkötter.

In the 2018 Nature Communications paper that is cited, the section "Impacts on physical, natural, and humans systems" discusses these issues. Here we've changed the verbiage to:

L444-445

> …can create challenging social and political environments stemming from the associated economic impacts.

As for the following sentence ([Former] 418-420), the intended meaning is fairly clear but the wording could be improved.

L446-448 Changed to:

> In the future, we should anticipate these ecosystem shifts as MHWs are expected to persist (Xu et al. 2021) and the atmospheric pressure systems associated with extreme events will increase in frequency (Giamalaki et al. 2021).

Figure 7d, 8d unit should be nM?

Yes. Corrected

Figure 9 unit needs a space between mg and m-3

Corrected

Figure 9 caption: There are a bunch of details about this Figure that are not really explained in the caption: the meaning of the vertical bars (probably mean, but needs to be stated, and panel (b) is different from the other 3), the vertical position of the symbols (arbitrary, but again should be stated), and the meaning of the symbol colours (obvious from the positions, but in this case is having two colours even necessary?) And there appear to be more years than there are symbols.

The new caption will read:

Observed and modeled summer (May–Aug) Chl (mg m$^{-3}$) contained in the large (left) and small (right) phytoplankton size fraction in two regions: (a,b) Alaska Gyre and (c,d) North Pacific Transition Zone. Model data are shown as normalized probability density functions for the MHW composite (red) and the climatology (gray) with the mean of each shown as a vertical bar on the x-axis. Observations from the six OSP cruises in the Alaska Gyre are shown as symbols on panels a and b for large (purple) and small (blue) phytoplankton respectively, at y=0.02. The data for non MHW years in 2000, 2001, 2008, and 2018 are shown as filled circles while data from the anomalous 2015 warm blob and 2013 volcanic eruption are shown by a star and a hollow circle . See also method Section 2.4.

---

## Author Response (AR2)

Reviewer comments in blue

Author comments in black

We thank the editor and both reviewers for their detailed and helpful feedback. The comments were very useful at improving the clarity of our paper and strengthened our results. In particular, we have added to and clarified the discussion of the significance of our results, adding a new section 2.2. We hope that the current version is acceptable for publication in Biogeosciences. Line by line responses to the reviewers are detailed below.

Referee #1

These authors have mostly done a good job of addressing the previous reviews. Some aspects of the statistical analysis are inadequately explained, and there are a number of ambiguities in the figure captions /legends.

Main points:

(1) I applaud the authors' efforts to add tests of statistical significance, but the description of what was done remains incomplete and unconvincing. The description on 87-94 is too terse to be useful. How exactly we get from what is described here to a significance level of 0.03 (253), 0.04 (330), or 0.001(331) is quite mysterious. Let's consider Figure 7b for example: there are 9 MHW years, and the Large Phytoplankton anomaly is negative in all of them. So it would certainly seem plausible that the probability of that occurring by chance is <5% or <1%. But one can not tell from the present text how that was demonstrated. We need more detail e.g., about the effective sample sizes. If we have 63 years of monthly data, but we are considering only a seasonal mean anomaly for the MHW years, then N2=9 and N1= at most 63. If we are considering monthly data then 63*12 is almost certainly an inflated estimate of N1 because of autocorrelation within the time series (e.g., 10.1175/1520-0442(1997)010<2147). If we are using annual or seasonal mean data then autocorrelation should be weak and N1~63 will not be too far off. In any case, whatever assumptions the authors are making, they need to spell them out in more detail.
We thank the reviewer for emphasizing the need for clear statistics. We have decided to include a separate section to discuss our statistical method. See section 2.2.

In response to this specific comment, we have used the annual anomaly (or seasonal mean anomaly as specified) for MHW (N=9) and non-MHW years (N=53), as now described in section 2.2.

(2) Two issues that are partially unresolved from the previous review (there are few minor ones as well, see below) are the description of iron limitation of small phytoplankton and question of cell-size-dependence of carbon-to-chlorophyll ratios. I still find the text ambiguous regarding whether the lack of iron-limitation of small phytoplankton is an emergent property or is programmed into the model a priori (e.g., 118, 283).

We have rewritten this part of the methods section. An iron deficit term is calculated for all size-classes; however, model results show it is only the "limiting nutrient" for large phytoplankton in this study. L131

"For macronutrients (e.g. nitrogen, phosphate), limitation factors are calculated using saturating kinetics while for iron, an internal iron deficiency term is calculated based on an internal cell quota (see supplementary materials in Stock 2014 for details). These limitation factors are output from the model as a number between zero and one, with zero indicating complete limitation, i.e. no phytoplankton growth, with the lowest value considered the limiting nutrient (Liebig 1840). Notably, in the study regions iron is only a limiting nutrient for large phytoplankton."

And the repeated claims that large phytoplankton have a systematically higher Chl:C ratio (e.g., 264, 290, 323) are unconvincing. Chl:C is function of temperature, irradiance, and cell nutrient status. There may be taxonomic differences, but they are generally much smaller than variation with environmental conditions. The repeated invocations of this idea sound glib and superficial and explain very little.

On L140-142 we specify how the simulated Chl:C ratio is calculated within the model as a function of temperature, light and nutrients, in agreement with the reviewer. When we reference the higher Chl:C ratio of large phytoplankton, however, we are referencing the model output Chl:C ratio for the 2 study regions. On L283 we specify what that ratio is for both size classes in the NPTZ.

"Seasonal Chl largely follows the large phytoplankton production due to a higher Chl:C ratio for large phytoplankton (0.022 vs. 0.014) simulated in this region (Geider et al. 1997; Stock et al. 2020)."

We have also added the simulated Chl:C to the AG section (L343-345) for completeness.

"The negative Chl anomaly that starts in the spring (April) is due to the decreased large phytoplankton, which have a higher simulated Chl:C (0.027), offset by the increased small phytoplankton (Chl:C = 0.016) production anomaly"

For clarity, we have added "simulated" to our references to the Chl:C ratio to make it clear that we are here talking about the modeled ratio.

(3) I think section 3.3 is too long and contains some superfluous text (e.g., 259-271). If this material is important enough to keep in the Results then the data are important enough not to be relegated to Supplementary. I would move this to Discussion or delete.

We have trimmed down that section of text to focus on the need to understand the seasonal cycle before discussing the MHW anomalies. However, these figures are best suited to the supplementary material to keep the main text streamlined.

Some details:

14 change "Yet" to "However" (see also 254, 350, 363)
Changed to "However" in all instances

33 delete "in the Northeast Pacific"
Removed

37 change "and/or" to "or"
L36 Change to "or"

51-52 "surface Chl alone provides little information on food web changes beyond primary production" Actually it doesn't necessarily tell us anything about production, just biomass.
L52 Yes, we have deleted this comment.

60 Polovina not in ref list
Thank you. Reference has been added.

111 "Tropics" should be "Trophics"
Yes, corrected.

113 not sure what the stray "12" means
Typo removed

140 "Mt. Pavlof eruption" is there a literature reference for this?
L158 Reference to Waythomas et al., 2014 added

142 "(2018)" something missing here
"2018" removed

145 italicize taxonomic names
L165 *Synechococcus* is now italicized

187 "west of P8 (P4–P20" P4 is east of P8
L208 Changed to "west of P4 (P4-P20)

188 "climatological" maybe not best word here
L209 Rephrased to "despite a model bias toward lower climatological surface nitrate"

190 "due to a lack of data prior to 2011" figure shows data before 2011
Yes, this statement should have been removed in the last iteration. It has been removed now.

203 add "North" before "American"
L224 Added

210 "intrusion of the nitrate-depleted region from the south into the NPTZ" expansion of the nitrate-depleted region northward into the NPTZ
L231 Corrected as suggested

217-220 I would change "the temporally variable, high-nitrate near-shore region" to "the highly variable but generally nutrient-rich near-shore region" and "the depleted nitrate region" to "the seasonally nitrate-depleted region"
L239-240 Corrected as suggested

221 change "Observations show" to "Ship-based Line P observations show"
L242 Corrected as suggested

223 delete "lateral"
Deleted

244 micromolar should be nanomolar?
L266 Yes, units corrected to nM

246 change "primary production" to "phytoplankton production"
L268 Changed to "phytoplankton"

275 change "dropping" to "declining"
L295 Changed to "declining"

281 "starts nearly a month earlier (early April vs late April)" how do they know this if working with monthly data?
The onset of nitrogen limitation is indicated by the intersection of the iron and nitrogen limitation shown in figure 8h, referenced immediately prior to this sentence. This sentence has been changed to the following:

L300 "the onset of nitrogen limitation, which occurs when the nitrogen limitation factor (dotted red line) intersects and drops below the iron limitation factor (dotted blue line), happens nearly a month earlier (early April) than the climatology (solid lines, late April) and the limitation factor is significantly lower"

284 something missing after "-2 mmol/m^2/d"
L303 "for large phytoplankton" added

329 "andwithin"
L351 Space added

339 "further supports this shifting of the phytoplankton assemblage" further supports the hypothesis of a shift in the phytoplankton assemblage
L362 Corrected as suggested.

360 "Our results agree with this literature" Our results support these previous studies
L385 Corrected as suggested.

383 change "depleted nitrate" to "nitrate-depleted"
L407 Changed to nitrate-depleted

397 "three different nitrate ..." something missing here
Corrected to "three different nitrate fields"

400 " Note that for ... the year is also used." On the one hand this seems like unnecessary belaboring of the obvious. But I understand the impulse behind it, as reviewers are often dense and demand explication of things like this. Maybe it can be worked into the figure caption.
L421 We have opted to keep this sentence for clarity.

414-415 This sentence borders on tautology. Maybe just delete everything after "both biomes".

We have removed the clause after "both biomes" as suggested.

L442 Changed to reduced

L 453 "(Fig. 9d)" Added

L462 – 464 hanged to:
"Evidence of this shift has been observed in the AG during the "warm blob" (Peña et al. 2019) which found higher concentrations of cyanobacteria in the nitrate-depleted region of Line P. Further, the data presented in this paper show higher Chl *a* concentrations in the smaller size classes at OSP (Sect. 3.3, 3.4)."

L470 Changed to "recur"

Caption has been updated to include box description (which is the box averaged in panel a, as assumed). All contours in b-e are colored dark gray, distinct from the Argo floats, and are the same values for SST panels (b,c) and chlorophyll panels (d,e). Text reference has been corrected to "Fig 1. d & e)."

Caption has been changed to correctly reference panels. Right side panels were incorrectly showing a similar analysis from the WOA. The WOA figures have been moved to the Supplementary Info and replaced with the Model sampling as described.

Caption changed to reference station map in Figure 1. Caption in figure 1 has been updated to explicitly mention the numbering of Line P stations.

"Line P (P1-P26) shown as yellow circles at every fourth Line P station with the black star denoting P26, also known as ocean station papa (OSP, 50° N, 145° W)."

"Boundary" has been replaced by "contour" in caption and throughout (L…).

Figure 5 - "boxes for AG and NPTZ are shown as described in Fig 1" Actually, Figure 1 has only one box and it is not explicitly defined.
Descriptions of the two study zones shown as boxes has been added to caption.

Figure 6 - MLD, POC not defined. Iron should be in nM? Change "Modeled composite of the 9 marine heatwave anomalies" to "Modeled composite anomaly of the 9 marine heatwaves".
MLD and POC now spelled out in the caption. Units for iron corrected to nM.
"Modeled composite of the 9 marine heatwave anomalies" changed to "Modeled composite anomaly of the 9 marine heatwaves" as suggested.

Figure 7 - The abbreviations "Lg" and "Sm" do not appear in the figure itself, and nowhere else in the text, so I would say they are expendable. Since the AG and NPTZ data are shown on separate panels, why does each warm composite need its own color?
Lg and Sm abbreviations have been dropped. We agree distinct coloring in this figure isn't necessary, but it follows the coloring of the boxes in Figures 5, 6 & 10 for continuity.

Figure 8/9 - Again the averaging box is misspecified: the box in Figure 1 is not black and its boundaries don't fit what is stated here. Also the y axis units do not appear to be stated anywhere. And there is something wrong with the legend to panel (h): I think "Mod MHW climatology" is not correct. And the main legend has a solid line for individual years while dashed lines are actually used.
Figures 8, 9 now correctly reference the black box representing the NPTZ and the red box representing the AG described in Figure 5. Y-axis labels have been added to all variables.
Legend in panel h has replaced "Mod MHW climatology" with "Mod climatology"

Figure 10 - Definitions of regions need to be stated (e.g., by reference to Figure 5 once its problems are fixed). Meaning of vertical bars still not explained.
Reference to Fig 5 added as well as the meaning of the vertical bars.

Figure 11 - y axis label should be anomaly in multiples of sigma?
I believe this comment references Figure 7? We have changed "units of sigma" to "multiples of sigma"

Maybe change "winter supply" to "winter inventory".
Caption for Fig 11 changed as suggested.

Terminology:

I have doubts about the usage of "Chl" as synonymous with "chlorophyll". When this abbreviation is defined, it refers to satellite chlorophyll (31). But then it is used to refer to field

measurements (e.g., 222), and there are several references to "Chl a". While oceanographers often use "chlorophyll" and "chlorophyll a" interchangeably, chlorophylls are a family of pigments of which chlorophyll a is only the most common (e.g., 101/m101p307). Satellite chlorophyll is an operationally defined quantity calculated from outgoing shortwave radiation at the top of the atmosphere, and implicitly includes absorption by other pigments. I think "Chl" should be reserved for satellite chlorophyll, and "chlorophyll" spelled out where it refers to field (bottle) data or specific pigments.

We agree with the reviewer and have changed all instances of chlorophyll to the full spelling "chlorophyll" except when referring specifically to Chl a, which is introduced in section 2.5, L155.

It is customary to choose a discrete significance threshold like 0.01 or 0.05. When I see a number like 0.03 (253) it makes me think the authors just chose the smallest value that all of their data exceeded. Not that this is wrong necessarily, or that it implies the test is artificial (but see my major comments above). But the convention is to e.g., use P<0.05 if 0.025<p<0.05.

In section 2.2 we have selected $p < 0.05$ to be our threshold for when the MHW years differ significantly from non-MHW years. This value is used throughout the text.

"nitrate limitation" still appears in several places (e.g., 15, 375)

Changed to "nutrient" limitation on lines 15, 375, 385, 386

"sigma" is sometime spelled out, sometimes a symbol

Changed to the symbol $\sigma$ throughout.

I don't like the terms "subtropical-like" and "subpolar-like". If the outstanding characteristic of the subtropical biome is that nutrients are depleted year round (371), why does a mid-latitude seasonal convection/stratification regime become "subtropical-like" just because seasonal drawdown becomes a bit more widespread or intense? And the use of the word "mode" (e.g., 310-311) is not consistent with its usual meaning.

Here we are specifically referencing the bimodal nature of the NPTZ chlorophyll distribution shown in Figure 10c & 10d. For both size-classes there is one mode that is consistent with the distribution shown for the subpolar AG, which we describe in lines L301-303. We now refer to this as a "high-chlorophyll mode", similar to the distribution of the subpolar gyre. The other mode is consistent with the subtropical gyre chlorophyll distribution which has a near-zero mean (L303-305) and we now refer to this as the "low-chlorophyll mode".

In in this section, we have changed the references from "peaks" to "modes" for added clarity.

Formatting:

Multiple references within a parenthesis should be arranged either alphabetically or chronologically; pick one and apply it consistently. This paper uses both, and sometimes neither (e.g., 60).

We have modified this using chronological order.

The figures are not cited in order in the text (e.g., 90).

Corrected

Always leave a space between a number and its unit (e.g., 228)
Corrected
* * *
Referee #2

This paper focuses on quantifying and explaining ecosystem impacts of marine heat waves in the NE Pacific, using a modelling approach supported by observational comparisons. The focus is on anomalies in chlorophyll and phytoplankton and their drivers. The topic is of high interest to the community. Some statistical tests for significance have been included in the revised manuscript, but in many places the significance is still unclear. This makes the conclusions of the paper unconvincing. The reader is left wondering whether there is any significant difference between the collection of MHW-years vs. normal years besides the large:small phytoplankton ratio. The most novel aspect of this paper is its consideration of the collection of MHW years, as previous work has investigated the 2014-15 event in some detail, so clarity in what is significantly different between the two populations of years is important. A few of my other previous comments have also not yet been fully addressed or revisions not fully incorporated throughout the paper.

The paper presents two new tests for significance: the Welch's T-test and a test for whether the mean of the MHW years falls outside the standard deviation of the climatology.
• For both tests, it's not clear whether the population that the MHW years are being compared to includes those years or not. I would recommend comparing the MHW years to the collection of years that are not MHW years. The text discusses comparison to climatology which would presumably include all years, MHW and non-MHW. Suggest adding statistics section in the methods and clarifying what tests were used and how they were done. In some cases, the conclusions of these two tests conflict with each other. For example, the p-value from the T-test suggests that lower small phytoplankton production, large phytoplankton production, and zooplankton production in the NPTZ during MHWs is significant but Figure 7 shows that the MHW mean is within 1 standard deviation of the climatology variability. I think it would be most straightforward to choose a single metric for significance and apply this across the board. Also, present an argument for why this is the best metric.
It is correct that this paper initially compared MHW years to the climatology, which did include all years.

We have updated our methods (Section 2.2) to now use the Kolmogorov-Smirnov (K-S) test instead of the Welch's T-test to evaluate ,using a significance threshold of p-value < 0.05, this test enables us to conclude that the ecosystem changes are a result of the MHW events.

Following this suggestion, we have updated our significance analysis so that MHW years are now compared to non-MHW years.

The comparison of MHW anomalies to the interannual standard deviation, is however not used to evaluate significance but to quantify the amplitude of the perturbation. As we show in Fig 7, some changes tied to MHW are significant (K-S T-test p-value< 0.05) but small (amplitude lower than 1 std). This suggests that there are sources of interannual variability that contribute more than MHWs (e.g., ENSO ref). Following the reviewer's suggestion, we have clarified our method by including a new, separate section (Methods 2.2) to discuss our statistical method. In there, we also added a sentence that explains the use of the standard deviation and modified the text in result/discussion to clarify.

There are multiple places in the paper where significance is not discussed at all when differences between all MHW and non-MHW are presented or where apparent differences are presented, discussed, and interpreted but then shown to be insignificant almost as a caveat. Where differences between MHW and non-MHW years are not significant, I don't think they should be presented and discussed at all, except to say that they are not significantly different. Below, I point out areas where statements of significance are required or removal of discussion of differences if they are not significant.

When it was determined that differences were not significant, we have either removed the quantification of the change, unless relevant to discussion.
e.g. L294, decrease in spring large phytoplankton production has been removed.

• Lines 209-213, 629-630: differences in the northward displacement of the 2 uM nitrate boundary

The argument we make here is qualitative. We have updated figure 5 to show that during MHW this limit is located north of the climatology.

For all other instances below, we have added "significant" or clarified the K-S's t-test p values in parenthesis (following the statistical methods description in section 2.2).

• Lines 167-168: differences in chlorophyll in each region
• Lines 240-244: differences in NPTZ chlorophyll, mixed layer depth, winter surface nitrate, and iron
• Lines 276-284: differences in NPTZ small phytoplankton production late winter / early spring, zooplankton production through May, iron limitation factor Jan-Apr, onset of nitrate limitation, nitrate limitation factor in ?late spring / summer?, small phytoplankton in June, total zooplankton production in June.
• Line 288: differences in NPTZ annual mean surface Chl anomaly
• Lines 303-310: differences in NPTZ large and small phytoplankton chl, % area of NPTZ with low large phytoplankton chlorophyll, % area of NPTZ with low small phytoplankton chlorophyll
• Line 320: differences in AG small phytoplankton production. The text later in this paragraph suggests this difference is not significant, in which case I don't think it should be presented and discussed.
• Lines 343-345: differences in AG large phytoplankton chlorophyll
• Lines 361-363: differences in AG and NPTZ nitrate and iron

• Lines 375-376: differences in NPTZ nitrate limitation factors
• Lines 426-432: differences in AG large phytoplankton production, annual small phytoplankton production

Treatment of the large:small phytoplankton ratio. Sometimes special statistics are needed when dealing with a ratio, because the variance may not be normally distributed in the ratio even when it is in the individual numerator and denominator. Please consider whether you can perform the same significance tests on the ratio.

To address this concern, we decided to do our statistical analysis of all variables between the non-MHW years and MHW years using a two-sample Kolmogorov-Smirnov test, which does not require a normal distribution or equal variances between the two datasets. We make note of this in section 2.2.

Comment from previous review: In general, the iron concentrations in the model (Figure S2) seem very high with modelled values around 100-150 nM iron in winter, whereas typical values for dissolved Fe in the region appear to be < 1 nM (see https://doi.org/10.1016/j.marchem.2015.04.004 for example).

This difference raises concerns that the model does not handle iron well in this region. Suggest that the authors explicitly discuss the model's simulation of iron, including comparison to observations, and the biases in that, especially since some of their conclusions rest on changing iron limitation.

This unit was incorrectly labeled, as noted by Reviewer 1. Surface iron concentrations throughout the paper are in units of nM (not μM). We have made sure this unit is correctly given in figures 6, 8, 9, S2, S3, S4 as well as correctly identified in the text L266. This corrects the model values to <1 nM (0.1 – 0.15 nM) throughout the region, in agreement with the cited reference.

How are the limitation factors defined? The authors have included a new statement that "limitation factors are output from the model as a number between zero and one, with zero indicating complete limitation, i.e. no phytoplankton growth". However, this does not clarify how the factor is calculated. Are these limitation factors the Michaelis-Menton values? If so, it would be useful to include the equation and limitation coefficients assumed for each species (in the supplemental material would be fine for this with a reference in the main text).

We have not included the equations as that is previously published in detail in Stock 2014. However, we have rewritten this section of the methods to include more details and clarify the equation number to look for in Stock et al L131-135.

"For macronutrients (e.g. nitrogen, phosphate), limitation factors are calculated using saturating kinetics while for iron, an internal iron deficiency term is calculated based on an internal cell quota (see supplementary materials in Stock 2014 for details). These limitation factors are output from the model as a number between zero and one, with zero indicating complete limitation, i.e. no phytoplankton growth, with the lowest value considered the limiting nutrient (Liebig 1845). Notably, in the study regions iron is only a limiting nutrient for large phytoplankton."

Minor suggestions:

Lines 100 and 190-191: Author response states that data from earlier years has now been obtained and incorporated into the paper. Revise text accordingly.
L114 Thank you, this oversight has been corrected and the references to the missing data have been removed.

Line 113: Phytoplankton12 ?
Typo removed

Lines 191-192: Figure 1 doesn't really support this statement. Modelled and satellite chlorophyll shown are for four years, not just the 2014-15 anomaly. The chlorophyll decline shown in Figure 1 is much less than the stated 0.3 mg m-3 and is found a lot farther south than Line P.
Reference to Fig. 1 has been removed

Lines 254-255: Confusing wording. "for all events" implies that this is true of each event, when the sentence refers to the composite.
Changed to "for the composite"

Figure 1: Suggest including the same contours in 1c as in 1b. Colour bar and axis labels overlap in 1c. Caption should specify what grey boxes indicate. Line 606: BGC-Argo floats with nitrate sensors (not all Argo floats).
Contours in 1c have been specified to match 1b. Color bars shifted. Caption updated as suggested.

Figure 3: The text around lines 173-175 say that the right hand side of this figure shows model data, but this part of the figure is labeled WOA – a climatology. High temperatures and low nitrate concentrations associated with the Blob are not visible in the WOA labeled panels.
Yes, this is an error. The WOA profile has been moved to the supplementary material and Figure 3 now appropriately has model date shown on the right-hand side, as described in lines 173-175.

Indeed, figure S2 showing the WOA profiles does not have the high temperature and nitrate concentrations associated with the warm blob because the sampled WOA nitrate field is climatological. However, the apparent shoaling of the thermocline and nitracline that occurs in 2015 is present, indicating that these features are not a result of the warm blob. Instead, this supports our argument that float #5904125 sampled a new water mass due to its northwest trajectory across the spatial nitrate gradient.

Figure 4: The lack of Line P chlorophyll observations near the coast seems incorrect. These regions are sampled on nearly every Line P cruise. The lack of surface salinity data also seems incorrect, particularly where there is temperature data.
This was an artifact of our colorbar selection. In the updated Figure 4, both salinity and chlorophyll show full coverage at the coast.

Figure 5: The boxes in this figure are not the same as Figure 1, and so should be described in the caption. Adjust caption for Figure 6 for the same reason.
Figure 5 caption has been changed to include box descriptions.

"The North Pacific Transition Zone (NPTZ, 39°−45°N and 160°−135°W) shown as a black box and the Alaskan Gyre (AG, 48°−54°N and 160°−145°W). Line P stations and OSP are shown as described in Fig 1."

Figure 6 caption now references the description in Figure 5.
"Line P, OSP as shown in Fig. 1; boxes for AG and NPTZ are shown as described in Fig. 5."

Figure 8: units for the values on the y-axis should be included in the figure or at least the caption. First (h) in the caption should be (g).
Units now provided along y-axes in Figure 8 (and Figure 9).

Figure 10: Caption should state what the vertical lines are.
Caption edited to include

"The mean of each is shown as a short vertical line on the x-axis (red, black respectively)"

---

## Author Response (AR3)

Reviewer comments in blue

Author comments in black

We thank the editor and both reviewers for all the feedback that was provided to strengthen and clarify the work in our manuscript. The additional minor and technical comments have been addressed and corrected. We look forward to publication in Biogeosciences. Line by line responses to the reviewers are detailed below.

Referee #1

This revision fully addresses my previous concerns about the statistical significance of the results. I find the conclusions of the paper much more convincing now. I have a couple of very minor additional suggestions or corrections. Line numbers are from the manuscript not the tracked changes version.

In most cases throughout the paper, the p value is just listed as p<0.05. Where convenient (for example where just one test is being discussed), I suggest listing the actual p value. The amount below 0.05 is informative.

In a prior version of the draft we used more specific p-values, however we decided it was simpler for the reader to use a consistent threshold value of $p < 0.05$.

Line 193. Fig. 3 a-c should be Fig. 3 a, c, e.

Corrected

Line 195. Fig. 3 d-f should be Fig. 3 b, d, f.

Corrected

Line 233 and Figure 5. I really like the change to Fig. 5 in adding the individual years rather than showing the MHW composite. The text near Line 233 needs to be revised accordingly.

Corrected  from "the 2 $\mu$M nitrate contour during marine heatwaves (thick dashed line)" to "the 2 $\mu$M nitrate contour during marine heatwaves (thin dashed lines)"

Line 298 – 8g should be 8e?
Corrected

Line 635 – stray character
Character deleted
* * *
Referee #2
This paper is mostly finished subject to some minor technical corrections. Most of the errors that I pointed out in the figure legends/captions have been corrected.

I am a bit concerned by this sentence in the Abstract: "Prior work showed that these events, including the 2014–2015 "warm blob," are associated with widespread surface nutrient declines in the subpolar Alaska Gyre (AG) and the North Pacific Transition Zone (NPTZ), but reduced concentrations in the NPTZ only." Does this make sense? How can surface 'nutrient' decline but nutrient concentration does not? The implication is that the latter refers to concentration of something else. Maybe there is a word or a symbol missing after "reduced". It looks like there is some extra space; maybe a special character just failed to render in the PDF.

Corrected to read: "reduced chlorophyll concentrations in the NPTZ only"

2.2 Statistics

On 91-100 specify whether the overall "warming" trend removed was a linear trend (probably, but good to be clear).

"linear" added to decription of warming trend

I still think it is necessary to discuss autocorrelation: clearly these authors do not think it is important and quite possibly they are correct. I would just add a single sentence: "As we used annual or seasonal mean data, we assume that autocorrelation is negligible and calculate significance based on the total number of years in the time series."

Sentence added as suggested

L135 Liebig 1840 is not an appropriate reference here. Liebig's Law refers to yield, not growth rate (see e.g., Droop 1983 Bot. Mar. 26: 99; Cullen 1991 L+O 36: 1578; Litchman 2007 in Falkowski and Knoll, eds, "Evolution of Primary Producers in the Sea"; https://www.sciencedirect.com/topics/agricultural-and-biological-sciences/liebigs-law-of-the-minimum)

Reference changed to Droop 1983 and added to bibliography

Figures 1 + 4: the contour lines still look black to me (stated in caption to be gray).

Changed to black in Fig 1, but the contour is gray in Fig 4.

Figure 5: description of the black and red boxes is still incomplete.

Specified that AG is a red box.

Figure 6: "organic" misspelled

Corrected

Figures 8 + 9: There still seems to be a discrepancy between the legends ("Mod MHW composite" in one case and "MHW. MHW composite" in the other), and between the linestyle shown in the legend for individual years and that actually used in the plots.

The legend on figure 8 has been corrected to read "Mod. MHW composite" and the individual years line styles in the legends of Figures 8 & 9 have been corrected.